# Graph Diffusion for Robust Multi-Agent Coordination

Xianghua Zeng [1]   Hang Su[*][2]   Zhengyi Wang [2]   Zhiyuan Lin [2]

## Abstract

Offline multi-agent reinforcement learning (MARL) struggles to estimate out-of-distribution states or actions due to the absence of real-time interactions with the environment. Although diffusion models have shown promising potential in addressing these challenges, they primarily apply independent diffusion to the historical trajectories of individual agents, which overlooks the crucial dynamics in multi-agent coordination and limits the policy robustness in dynamic environments. In this paper, we propose **MCGD**, a novel **M**ulti-agent **C**oordination framework based on **G**raph **D**iffusion models to improve the effectiveness and robustness of collaborative policies. Specifically, we construct a sparse coordination graph with continuous node attributes and discrete edge attributes to identify the underlying multi-agent dynamics effectively. We then derive the transition probabilities between edge categories and present adaptive categorical diffusion to model the structure diversity of inter-agent coordination. According to the coordination structure, we define the neighbor-dependent forward noise and design anisotropic diffusion to increase the action diversity of each agent. Extensive experiments across various multi-agent environments demonstrate that MCGD significantly outperforms existing state-of-the-art baselines in coordination performance and exhibits superior robustness to dynamic environmental changes.

[1]State Key Laboratory of Software Development Environment, Beihang University, Beijing, China [2]Department of Computer Science and Technology, Institute for AI, BNRist Center, Tsinghua-Bosch Joint ML Center, Tsinghua University, Beijing, China. Correspondence to: Hang Su <suhangss@tsinghua.edu.cn>.

*Proceedings of the $42^{nd}$ International Conference on Machine Learning*, Vancouver, Canada. PMLR 267, 2025. Copyright 2025 by the author(s).

## 1. Introduction

Offline Multi-Agent Reinforcement Learning (MARL) enables policy learning from pre-collected datasets, circumventing the need for real-time interactions with the environment (Lange et al., 2012; Levine et al., 2020). This approach is essential in scenarios where real-time interactions are expensive, unsafe, or infeasible, such as robotics in hazardous environments or autonomous systems (Barde et al., 2024; Wang et al., 2024). Offline MARL offers a path for deploying intelligent agents without requiring continuous interaction, but it introduces significant challenges.

The primary challenges in offline MARL arise from the absence of real-time feedback, which limits the ability to adapt policies dynamically based on ongoing interactions with the environment (Matsunaga et al., 2023). First, generalization to unseen states and actions is difficult because policies are trained on limited data, which may not fully capture the diversity of real-world scenarios. In multi-agent settings, this issue is compounded by the complexity of agent interactions. Second, out-of-distribution (OOD) actions and states present another challenge, as offline methods often struggle to handle situations not represented in the training data. This can lead to unreliable or unsafe behaviors due to the model's inability to properly extrapolate from the limited data available (Kumar et al., 2019; Fujimoto et al., 2019).

Recent advancements have integrated diffusion models (Song & Ermon, 2019; Ho et al., 2020) into both single-agent and multi-agent offline reinforcement learning (RL) to improve policy stability and performance. In single-agent offline RL, diffusion models address issues like overestimation bias by modeling the full distribution over actions, leading to more robust value function estimates (Janner et al., 2022; Ajay et al., 2022). This probabilistic approach helps mitigate common challenges in high-dimensional state spaces by capturing complex dependencies between states and actions, offering a more stable alternative to traditional Q-learning. For multi-agent offline RL, diffusion models like MADIFF (Zhu et al., 2023) extend this framework to model complex agent interactions, leveraging diffusion processes to simulate cooperative dynamics within multiple interacting agents. Additionally, methods such as EAQ (Oh et al.) enhance the training process by incorporating the Q-total function into the diffusion model, improving

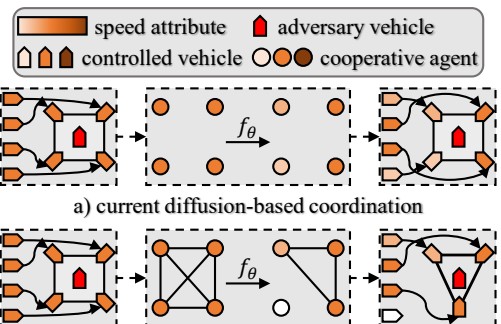

Figure 1: Comparison between current diffusion-based algorithms and our graph diffusion-based framework in an illustrative four-agent hunting scenario, focusing on dynamic changes in speed attributes and coordination structures.

the estimation of joint action values in multi-agent settings. DOM2 (Li et al., 2023), on the other hand, overcomes the conservatism of offline RL by enabling greater exploration of strategies, thus reducing the tendency to rely on suboptimal, overly cautious policies.

Nevertheless, these methods typically apply diffusion models independently to the historical trajectories of individual agents, overlooking the crucial coordination dynamics between agents. This approach limits policy robustness in dynamic multi-agent settings, as it adapts to changes in individual agent attributes but fails to capture the evolving coordination strategies that emerge as agents interact. Figure 1 illustrates this limitation using a four-agent collaborative hunting scenario, where four vehicles work together to capture an adversary. This scenario mimics real-world challenges in multi-agent systems, where agents must continuously coordinate and adapt to both internal and external changes. The hunting strategy, trained on fixed speed attributes from offline data, fails to generalize to real-time shifts, such as changes in agent speed or the sudden unavailability of an agent. When two agents' speeds change, the diffusion-based method can adjust the strategy through noise addition and denoising, as shown in Figure 1(a), allowing the task to proceed. However, when one agent unexpectedly becomes unavailable, the remaining agents cannot adapt their coordination structure, and the hunting task fails. This example highlights a critical challenge in offline MARL: the need for policies that not only adapt to individual agent attributes but also dynamically adjust to shifts in the coordination structures. In real-world applications, such changes in coordination are inevitable, and robust policies must be able to handle these shifts to ensure successful and efficient multi-agent collaboration.

In this work, we propose a novel generative framework **MCGD** for **M**ulti-agent **C**oordination based on **G**raph **D**iffusion models to enable diverse and adaptive collaborative policies, as illustrated in Figure 1 (b), with enhanced effectiveness and robustness in dynamic environments. Ini-

tially, to identify the collaborative dynamics, we construct a sparse graph with continuous nodes and discrete edges, retaining essential and eliminating ineffective multi-agent interactions. Next, we measure the observational differences to derive an edge transition matrix adapted to multi-agent behaviors, and present categorical diffusion to model the structure diversity in inter-agent coordination. For each agent, we define neighbor-dependent forward noise to capture the dynamic coordination structure and develop anisotropic diffusion to model the diversity in single-agent actions. Finally, extensive comparative evaluations on three well-established benchmarks-MPE, MAMuJoCo, and SMAC-demonstrate that our framework achieves superior coordination performance and robustness, significantly outperforming state-of-the-art baselines by up to 12.8% and 14.2%, respectively. Our contributions are summarized as follows:

• We propose the first graph diffusion model for multi-agent coordination with superior effectiveness and robustness in dynamic multi-agent environments.

• We present a categorical diffusion process to simulate transitions between edge categories, modeling the structure diversity in multi-agent coordination.

• We develop an anisotropic diffusion process incorporating neighbor-dependent forward noise to model the diversity in single-agent actions.

• Comparative evaluations across various challenging multi-agent environments demonstrate the significant advantages of our framework in coordination performance and policy robustness compared to state-of-the-art baselines.

## 2. Related Work

### 2.1. Offline Multi-agent Reinforcement Learning

Offline coordination is a significant hurdle that limits advancements and exploration within offline multi-agent reinforcement learning (MARL) (Zhu et al., 2023). To address this, recent studies (Chen et al., 2021; Yang et al., 2021) have extended single-agent offline algorithms to multi-agent scenarios through policy regularization. However, these extensions face persistent extrapolation errors (Fujimoto et al., 2019) in offline environments, which remain challenging to resolve fully. An alternative approach, the centralized training with decentralized execution (CTDE) paradigm (Oliehoek et al., 2008), has driven notable progress through algorithms such as QMIX (Rashid et al., 2020), MADDPG (Lowe et al., 2017), and MAPPO (Yu et al., 2022). Additionally, methods like MA-ICQ (Yang et al., 2021) and OMAR (Pan et al., 2022) have been developed to address distributional shifts in offline settings by adopting conservatism principles. Despite their success in reducing distributional errors, these conservatism-based methods restrict the flex-

ibility of agent coordination, limiting adaptability across varied scenarios. To enhance multi-agent coordination, diffusion models capable of capturing complex distributions have recently been introduced into offline MARL (Zhu et al., 2023; Li et al., 2023; Oh et al.). However, these algorithms independently apply diffusion models to the historical trajectories of individual agents, neglecting the inter-agent coordination structure, which reduces their robustness to environmental changes.

In response, this work introduces a graph diffusion-based multi-agent coordination framework to explicitly capture the coordination structure and utilize distinct diffusion processes for structure and action diversity, achieving superior adaptability and performance in dynamic environments.

### 2.2. Diffusion Models

Diffusion models (Sohl-Dickstein et al., 2015; Ho et al., 2020) have emerged as a powerful generative framework for modeling complex data distributions. They have shown significant success in continuous generation tasks such as image synthesis (Dhariwal & Nichol, 2021; Rombach et al., 2022), animation creation (Ho et al., 2022; Luo & Hu, 2021), and molecule design (Corso et al.; Trippe et al.). Recently, two distinct strategies have been proposed to adapt diffusion models for generating graphs with discrete structures (Austin et al., 2021; Hoogeboom et al., 2021). The first approach (Chen et al.; Hoogeboom et al., 2022) embeds graph data into a continuous space by adding Gaussian noise to the node features and adjacency matrix, enabling the model to learn the underlying graph distribution. An alternative strategy (Vignac et al.; Hua et al., 2024) avoids continuous perturbations, which are less effective at capturing the structural properties of graph data. Instead, it directly models categorical diffusion for discrete graph data by estimating the transition probabilities between different categories. However, existing graph diffusion models primarily focus on the independent diffusion of discrete attributes across nodes and edges. This limitation makes them unsuitable for adaptively modeling the diversity of discrete multi-agent structures and continuous single-agent actions in offline multi-agent reinforcement learning.

To address these challenges, we propose the first graph diffusion model for modeling multi-agent collaboration, which combines categorical diffusion for inter-agent structures with continuous diffusion for multi-agent actions, facilitating more effective and robust collaboration.

## 3. Preliminaries

In this section, we formally define the fundamental concepts and provide a summary of the primary notations, as detailed in Appendix 7.1.1.

### 3.1. Offline Multi-agent Reinforcement Learning

We model the fully cooperative multi-agent task as a decentralized partially observable Markov decision process (Dec-POMDP) (Oliehoek et al., 2016), described as a tuple $\mathcal{M} = \langle \mathcal{I}, \mathcal{S}, \mathcal{A}, \mathcal{P}, \Omega, \mathcal{O}, \mathcal{R}, \gamma \rangle$ with agents $\mathcal{I} = \{n_1, n_2, \ldots, n_{|\mathcal{I}|}\}$. In this context, $\mathcal{S}$ and $\mathcal{A}$ denote state and action spaces, respectively, and $\gamma \in [0, 1]$ is the discount factor. At each timestep $t$, every agent $n_i$ observes a local observation $o_i^t \in \Omega$ produced by the function $\mathcal{O}(o_i^t | S_t, a_i^t)$ and chooses an action $a_i^t \in \mathcal{A}$. All chosen actions form a joint action $A_t \in \mathcal{A}^{|\mathcal{I}|}$ and lead to a transition to the next global state $S_{t+1}$ according to the function $\mathcal{P}(S_{t+1}|S_t, A_t)$, resulting in a joint reward $r_t = \mathcal{R}(S_t, A_t)$. In offline settings, rather than interacting with the environment in real-time, we have access to a historical dataset $\mathcal{D}$ to learn multi-agent policies that maximize the discounted cumulative reward. The offline dataset $\mathcal{D}$ typically consists of multi-agent observation-action trajectories, where each trajectory $\boldsymbol{\tau} = [O_0, A_0, O_1, A_1, \ldots, O_T, A_T]$ includes the joint observations and actions at each time step.

### 3.2. Denoising Diffusion Probabilistic Model

The denoising diffusion probabilistic model (DDPM) (Ho et al., 2020) is a generative model that synthesizes continuous or discrete data through a forward noising process and a corresponding reverse denoising process.

In continuous diffusion, DDPM introduces Gaussian noise to the original data $X_0 \sim q(X)$ at each iteration $k$, perturbing it progressively as follows:

$$q(X_k|X_{k-1}) = \mathcal{N}(X_k; \sqrt{1 - \beta_k} X_{k-1}, \beta_k I), \quad (1)$$

where $I$ denotes the identify covariance matrix, and $\beta_k \in (0, 1)$ controls the scale of the Gaussian noise added at iteration $k$. In the reverse process, DDPM employs a trained, parameterized Gaussian transition kernel $p_\theta$ to iteratively denoise samples, gradually reconstructing the original data distribution from the noisy samples as follows:

$$p_\theta(X_{k-1}|X_k) = \mathcal{N}(X_{t-1}; \mu_\theta(X_k, k), \Sigma_\theta(X_k, k)), \quad (2)$$

where $\mu_\theta$ and $\Sigma_\theta$ represent the predicted average value and covariance matrix parameterized by $\theta$.

For discrete diffusion on data $X$ (e.g., graph structures), DDPM computes the transition probabilities between categories to replace the Gaussian noise, yielding the noisy data at iteration $k$ as follows:

$$q(X_k|X_{k-1}) = \text{cat}(X_k|X_{k-1}Q_{\beta_k}), \quad (3)$$

where $Q_{\beta_k}$ represents the transition matrix applied to the discrete data $X_{k-1}$ at iteration $k$, and $\text{cat}$ denotes the categorical distribution over possible categories of $X_k$. During

the denoising process, the posterior distribution $p_\theta$ is computed using the Bayes rule as follows:

$$p_\theta(X_{k-1}|X_k, X) \propto \text{cat}(X_k[Q_{\beta_k}]^T \odot X\overline{Q}_{\beta_{k-1}}), \quad (4)$$

where $\overline{Q}_{\beta_{k-1}} = Q_{\beta_1}Q_{\beta_2}\ldots Q_{\beta_{k-1}}$, $[Q_{\beta_k}]^T$ represents the transpose of the transition matrix $Q_{\beta_k}$, and $\odot$ denotes the Hadamard product.

# 4. Methodology

In this work, we propose the first graph-based diffusion framework for offline multi-agent reinforcement learning, including categorical diffusion on discrete edges and anisotropic diffusion on continuous nodes, to model the diversity of inter-agent coordination and single-agent actions, respectively. The proposed framework consists of three primary processes: forward noising, reverse denoising, and policy sampling, as detailed in Figure 2. Specifically, we perturb forward the nearest-neighbor coordination graph constructed from multi-agent historical trajectories, design the graph transformer network to reversely recover the clean attributes of discrete edges and continuous nodes, and employ the trained diffusion model to sample collaborative policies for multi-agent decentralized execution.

For clarity, we denote the timestep in an episode as $\{t\}_{t=1}^T$, the diffusion iteration as $\{k\}_{k=1}^K$, and the agent as $\{n_i\}_{i=1}^{|\mathcal{I}|}$.

## 4.1. Forward Noising Process

To capture the collaborative dynamics among agents, we identify the essential multi-agent interactions to construct the sparse coordination graph $G_t = (A_t, E_t)$ at timestep $t$. For each agent $n_i \in \mathcal{I}$, we establish undirected edges connecting it with its $k$ nearest neighbors defined by their observation difference, resulting in $k$-nn coordination graph. In the graph $G_t$, $A_t \in \mathbb{R}^{|\mathcal{I}| \times d}$ represents the $d$-dimensional continuous node attributes that encode the actions of each agent, while $E_t \in \{0,1\}^{|\mathcal{I}| \times |\mathcal{I}|}$ is a binary adjacency matrix where each entry indicates the presence (or absence) of an edge between two agents. For discrete action spaces, we represent each agent's action using a one-hot vector of dimension $d$, where $d$ is the number of discrete actions.

In this subsection, we present a graph diffusion process over the graph $G_t$ to simulate the dynamic multi-agent behavior when online coordination, detailed as follows:

$$q(G_k^t|G_{k-1}^t) = (\sqrt{1-\beta_k}A_{k-1}^t + \beta_k\epsilon, E_{k-1}^tQ_{\beta_k}), \quad (5)$$

where $\epsilon$ and $Q_{\beta_k}$ are the forward noise and transition matrix applied $A_{k-1}^t$ and $E_{k-1}^t$, respectively, at iteration $k$. To guarantee the efficiency and adaptivity of the forward noising process, three desirable properties are required:

• The transition matrix $Q_{\beta_k}$ should be adaptive to the multi-agent historical trajectories.

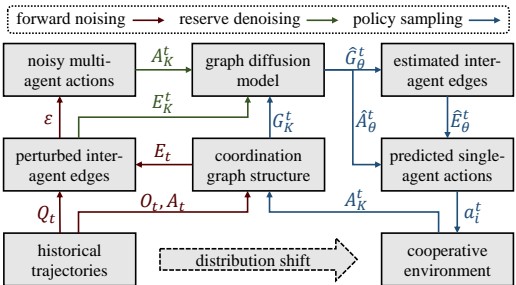

Figure 2: The overall architecture of MGCD, including forward noising process, reverse denoising process, and policy sampling process.

• The marginal distribution $q(G_K^t|G_t)$ should have a closed-form expression for efficient calculation.

• The Gaussian noise $\epsilon$ should be anisotropic and dependent on the inter-agent coordination $E_t$.

### 4.1.1. CATEGORICAL NOISING ON EDGES

Considering the relevance among multiple agents, we define an adaptive transition matrix between edge categories and present a categorical diffusion process on the discrete edges to model the diversity in inter-agent coordination.

To quantify the multi-agent relevance, for each pair of agents $n_i$ and $n_j$, we calculate the cosine similarity $c_{i,j}$ between their observations $o_i^t, o_j^t \in O_t$ as a measure of the likelihood of diffusion occurring between $n_i$ and $n_j$. Intuitively, a high similarity in the agents' observations reflects a strong alignment of their states and indicates the potential for substitutable coordination, thereby increasing the likelihood of diffusion between them. Leveraging this metric, we construct a similarity matrix $C \in \mathbb{R}^{|\mathcal{I}| \times |\mathcal{I}|}$ and derive the transition matrix $Q_t$ between edge categories as follows:

$$Q_t = e^{(C-D)t}, \quad (6)$$

where $D$ is the diagonal degree matrix of $C$, normalizing the similarity values. Theorem 4.1 establishes the rationality of the transition matrix $Q_t$ for categorical diffusion, confirming the necessary properties identified in prior studies (Yi et al., 2024; Shi et al.).

**Theorem 4.1.** *The transition matrix $Q_t$ satisfies the following beneficial properties of symmetry, additivity, locality, and convergence:*

$$[Q_t]^T = Q_t, \quad Q_{t_i+t_j} = Q_{t_i}Q_{t_j},$$
$$\lim_{t \to 0} Q_t = I, \quad \lim_{t \to \infty} Q_t = \frac{1}{|\mathcal{I}|}\mathbf{11}^T. \quad (7)$$

The detailed proof is provided in Appendix 7.2.1.

By incorporating the scale parameter $\beta_k$, the categorical noising of edge discrete $E_t$ at diffusion iteration $k$ is formal-

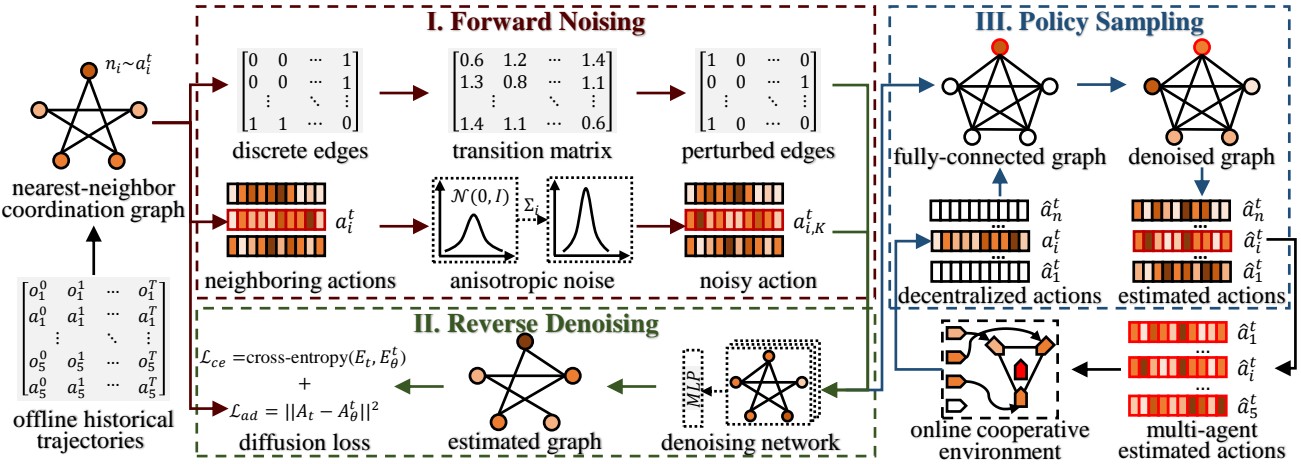

Figure 3: Detailed designs of our proposed graph diffusion-based coordination framework.

ized as follows:

$$q(E_k^t|E_{k-1}^t) = \text{cat}(E_k^t|E_{k-1}^tQ_{\beta_k}), \quad Q_{\beta_k} = e^{(C-D)\beta_k}, \tag{8}$$

where $\text{cat}(\cdot)$ represents the categorical distribution, parameterized by the product $E_{k-1}^tQ_{\beta_k}$. Utilizing the symmetry and additivity properties in Theorem 4.1, we derive the closed-form expression for categorical noising on $E_t$:

$$q(E_K^t|E_t) = \text{cat}(E_K^t|E_t\overline{Q}_{\beta_K}),$$
$$\overline{Q}_{\beta_K} = Q_{\beta_1}Q_{\beta_2}\ldots Q_{\beta_K} = e^{\sum_{k=1}^K[\beta_k(C-D)]}. \tag{9}$$

### 4.1.2. ANISOTROPIC NOISING ON NODES

Instead of the noising process on historical trajectories independent of inter-agent coordination, we define a forward Gaussian noise conditioned on neighboring agents' actions and then design an anisotropic diffusion on continuous nodes to model the diverse single-agent actions.

For each agent $n_i \in \mathcal{I}$, we encode the action information of its neighbors in the original coordination graph $G_t$ as $A_i^t \subset A_t$, and calculate the covariance matrix $\Sigma_i \in \mathbb{R}^{d \times d}$ of $A_i^t$ to characterize the anisotropy of its diffusion process. The forward noising over the single-agent action $a_i^t$ at iteration $k$ is formalized as follows:

$$q(a_{i,k}^t|a_{i,k-1}^t) = \mathcal{N}(a_{i,k}^t; \sqrt{1-\beta_k}a_{i,k-1}^t, \beta_k\Sigma_i). \tag{10}$$

Given the parameters $\alpha_k = 1 - \beta_k$ and $\overline{\alpha}_K = \prod_{k=1}^K \alpha_k$, the closed-form expression of the marginal distribution of $a_{i,K}^t$ is derived as follows:

$$q(a_{i,K}^t|a_i^t) = \mathcal{N}(a_{i,K}^t; \sqrt{\overline{\alpha}_K}a_i^t, (1-\overline{\alpha}_K)\Sigma_i). \tag{11}$$

The detailed proof is provided in Appendix 7.2.2.

By integrating the categorical diffusion over discrete edges and anisotropic diffusion over continuous actions, we derive

the closed-form expression for the marginal distribution $q(G_K^t|G_t)$ as follows:

$$q(G_K^t|G_t) = (\sqrt{\overline{\alpha}_K}A_t + (1-\overline{\alpha}_K)\epsilon, E_t\overline{Q}_K), \tag{12}$$

where $\epsilon$ deontes the multi-agent $d$-dimensional anisotropic Gaussian noise, with each component $\epsilon_i$ for agent $n_i$ drawn from $\mathcal{N}(0, \Sigma_i)$.

### 4.2. Reverse Denoising Process

In this subsection, we design a graph diffusion network to reverse denoise the perturbed coordination graph, with the goal of recovering the original attributes of inter-agent coordination and single-agent actions.

Given the edge attributes $E_k^t$ and $E_t$, we apply the Bayes rule to derive the posterior distribution of $E_{k-1}^t$ as follows:

$$q(E_{k-1}^t|E_k^t, E_t) \propto \text{cat}(E_{k-1}^t|E_k^tQ_{\beta_k} \odot E_t\overline{Q}_{\beta_{k-1}}), \tag{13}$$

where $\odot$ denotes the pairwise Hadamard product operation.

To predict the unperturbed edge attributes $E_t$, we employ a graph transformer network (Yun et al., 2019) to construct a graph denoising network $f_\theta$. This network takes as input the noisy graph $G_K^t = (A_K^t, E_K^t)$ and outputs the estimated edge attribute $\hat{E}_\theta^t$ and node attribute $\hat{A}_\theta^t$

To optimize the graph diffusion model parameterized by $\theta$, we employ the cross-entropy loss $\mathcal{L}_{ce}$ to quantify the discrepancy between the clean edge attribute $E_t$ and the predicted edge attribute $\hat{E}_\theta^t$ as follows:

$$\mathcal{L}_{ce} = \mathbb{E}_{(O_t, A_t) \in \tau} \sum_{i,j} \text{cross-entropy}(E_{i,j}^t, \hat{E}_{i,j}^t), \tag{14}$$

where $E_{i,j}^t$ and $\hat{E}_{i,j}^t$ denote the $(i,j)$ entry in the discrete attributes in $E_t$ and $\hat{E}_\theta^t$, respectively.

To further optimize the denoising network $f_\theta$, we compute the Euclidean distance between the clean node attribute $A_t$

and the predicted node attribute $\hat{A}_\theta^t$, and incorporate the Q-loss (Wang et al.) to define the anisotropic diffusion loss $\mathcal{L}_{ad}$, as follows:

$$\mathcal{L}_{ad} = \mathbb{E}_{(O_t, A_t) \in \tau} \sum_{i=1} \left[ ||a_i^t - \hat{a}_i^t||^2 - \lambda \mathcal{Q}_{\phi_i}(\overline{o}_i^t, \hat{a}_i^t) \right],$$
(15)

where $\hat{a}_i^t \in \hat{A}_t$ denotes the predicted action of agent $n_i \in \mathcal{I}$, $\overline{o}_i^t$ represents the average observation of agent $n_i$ and its neighbors, and $\lambda$ is the regularization coefficient.

To reduce model complexity and maintain scalability, we process each neighboring observation using a shared-parameter MLP, followed by a mean pooling operation over the resulting features. This replaces concatenation, which can significantly increase the parameter count as the number of neighbors grows. By limiting the neighbors to those with higher similarity, the averaging operation is performed over semantically similar features, thereby mitigating information loss.

The training details of our MCGD framework are provided in Appendix 7.1.2.

### 4.3. Policy Sampling Process

In this subsection, we design a policy sampling process based on the trained graph diffusion model to generate multi-agent collaborative behaviors, with the aim to progressively denoise the inter-agent coordination and multi-agent actions in a decentralized execution setting. The whole sampling process is summarized in Algorithm 1.

At each timestep $t$, each agent $n_i \in \mathcal{I}$ obtains its local observation $o_i^t$ from the environment and selects the optimal action $a_i^t$ according to its trained Q-value function $\mathcal{Q}_{\phi_i}^*$ (line 5 in Algorithm 1). Instead of the noise initialization used in prior methods (Li et al., 2023; Zhu et al., 2023), for each agent $n_i$, we generate $N$ random action samples from the continuous action space to construct a candidate set. We then evaluate each candidate using the trained Q-function $\mathcal{Q}_{\phi_i}^*$ and select the action with the highest Q-value, providing a more informed and value-guided sampling strategy. Since the optimal action is selected from a finite set of sampled candidates, this approach is naturally applicable to both discrete and continuous action spaces and does not require differentiability or closed-form maximization over actions. For the actions of other agents that are inaccessible, we replace them with a standard Gaussian noise of the same dimensionality, thereby forming the multi-agent actions $A_t$.

Using the actions $A_t$ as continuous node attributes, we initialize a fully connected coordination graph $G_i^t = (A_t, E_t)$ (line 6 in Algorithm 1), where the edge attribute $E_t$ is defined such that off-diagonal entries are set to 1. The attributes $E_t$ and $A_t$ are treated as the initial perturbed data $E_K^t$ and $A_K^t$ in the context of categorical and anisotropic

---

**Algorithm 1** MCGD Sampling Algorithm

1: **Input:** diffusion parameter $K$, graph diffusion model $f_{\theta^*}$, and Q-value function $\mathcal{Q}_{\phi_i}^*$ for each agent $n_i$
2: **Initialize:** initial observation $o_i^0$ for each agent $n_i$
3: **for** each timestep $t$ **do**
4:     **for** each agent $n_i$ **do**
5:         Select action $a_i^t$ based on $\mathcal{Q}_{\phi_i}^*$
6:         Construct the fully-connected graph $G_i^t$
7:         **for** each diffusion iteration $k$ **do**
8:             Predict the clean attributes $\hat{E}_\theta^t$ and $\hat{A}_\theta^t$ based on the denoising network $f_\theta$
9:             Sample the edge attributes $E_{k-1}^t$ from the posterior distribution $p_\theta(E_{k-1}^t | E_k^t, \hat{E}_\theta^t)$
10:         **end for**
11:         Execute the single-agent action $a_i^t \in \hat{A}_\theta^t$
12:     **end for**
13: **end for**

---

diffusion processes, respectively.

At each diffusion iteration $k$, we employ the denoising network $f_\theta$ to predict the denoised edge attribute $\hat{E}_\theta^t$ and node attribute $\hat{A}_\theta^t$ (line 8 in Algorithm 1). Using the edge attributes $\hat{E}_\theta^t$ and $E_k^t$, we derive the posterior distribution $p_\theta(E_{k-1}^t | E_k^t, E_0^t)$, from which we sample the edge attribute $E_{k-1}^t$ at iteration $k-1$ (line 9 in Algorithm 1), as follows:

$$E_{k-1}^t \sim \text{cat}(E_{k-1}^t | E_k^t Q_{\beta_k} \odot f_\theta(E_k^t, k)\overline{Q}_{\beta_{k-1}}). \quad (16)$$

Based on the sampled edge attribute $E_{k-1}^t$ and the predicted node attribute $\hat{A}_\theta^t$, we construct the perturbed coordination graph $G_{k-1}^t = (\hat{A}_\theta^t, E_{k-1}^t)$, which will be used in the next denoising iteration $k-1$. After the final denoising iteration, we extract the single-agent action $a_i^t$ from the final predicted node attribute $\hat{A}_\theta^t$ as the sampled action of agent $n_i \in \mathcal{I}$ at timestep $t$ (line 11 in Algorithm 1).

Under decentralized execution, each agent $n_i$ extracts its own action by retrieving the $i$-th row of $\hat{A}_\theta^t$, ensuring consistency with the fully decentralized setting. In discrete action spaces, the denoised continuous output is passed through a softmax layer, and the final discrete action is selected via an argmax operation over the corresponding softmax probabilities.

## 5. Evaluation

In this work, we conduct comparative experiments across different multi-agent environments, aiming to validate our method's ability to model the complex and diverse behaviors of cooperative agents. Specifically, we evaluate whether MCGD outperforms state-of-the-art baselines in coordination performance and demonstrates superior robustness to

Table 1: Comparison between MCGD and baselines on offline Expert or Good datasets across the MPE, MAMuJoCo, and SMAC benchmarks: "average value ± standard deviation". **Bold**: the best performance, underline: the second performance.

| Method | Expert MPE | | | Good MAMuJoCo | | | Good SMAC | | | |
|---|---|---|---|---|---|---|---|---|---|---|
| | Spread | Tag | World | 2halfcheetah | 2ant | 4ant | 3m | 2s3z | 5m6m | 8m |
| MA-ICQ | 79.2 ± 4.3 | 92.6 ± 15.5 | 83.5 ± 20.7 | 1735.2 ± 748.3 | 1186.2 ± 653.9 | 1214.9 ± 738.4 | 18.8 ± 0.6 | 19.6 ± 0.3 | 16.3 ± 0.9 | 19.6 ± 0.3 |
| MA-CQL | 74.1 ± 5.8 | 68.3 ± 13.2 | 57.7 ± 20.5 | 2722.8 ± 1022.6 | 1394.8 ± 604.3 | 1039.1 ± 617.5 | 19.6 ± 0.3 | 19.0 ± 0.8 | 13.8 ± 3.1 | 11.3 ± 6.1 |
| OMAR | 82.9 ± 2.4 | 97.9 ± 16.4 | 84.8 ± 21.0 | 2963.8 ± 410.5 | 1075.3 ± 374.1 | 954.8 ± 319.7 | 18.4 ± 0.2 | 18.8 ± 0.5 | 15.7 ± 0.3 | 16.2 ± 0.5 |
| MA-SfBC | 87.5 ± 7.3 | 77.4 ± 13.9 | 97.3 ± 19.1 | 2386.6 ± 440.3 | 1764.1 ± 457.4 | 1721.8 ± 392.3 | 19.1 ± 0.3 | 19.2 ± 0.3 | 15.2 ± 0.1 | 18.1 ± 0.3 |
| DOM2 | 88.7 ± 6.3 | 98.2 ± 14.4 | 99.5 ± 17.1 | 3676.8 ± 248.5 | 2187.4 ± 190.3 | 1836.2 ± 241.6 | 19.4 ± 0.2 | 19.0 ± 0.4 | 18.1 ± 0.7 | 18.2 ± 0.4 |
| MADIFF | 82.1 ± 5.9 | 103.0 ± 12.0 | 96.4 ± 13.7 | 3446.5 ± 213.3 | 2479.3 ± 105.8 | 2414.5 ± 128.3 | 19.6 ± 0.7 | 19.4 ± 0.1 | 18.0 ± 1.0 | 19.2 ± 0.1 |
| MCGD | **93.8 ± 2.7** | **109.6 ± 13.3** | **110.9 ± 11.5** | **3917.4 ± 193.7** | **2782.7 ± 203.9** | **2609.2 ± 165.2** | **22.1 ± 0.1** | **20.7 ± 0.1** | **18.9 ± 0.5** | **20.1 ± 0.2** |
| Abs.(%) Avg.↑ | 5.1(5.7) | 6.6(6.4) | 11.4(11.5) | 240.6(6.5) | 303.4(12.2) | 194.7(8.1) | 2.5(12.8) | 1.1(5.6) | 0.9(5.0) | 0.9(4.7) |

Table 2: Comparison between MCGD and baselines in shifted environments including MPE Spread, MPE Tag, MPE World, and MAMuJoCo 2halfcheetah, with dynamic changes in agent attributes and coordination structure: "average value ± standard deviation". **Bold**: the best performance in each graph, underline: the second performance.

| Method | MPE Spread | | MPE Tag | | MPE World | | MAMuJoCo 2halfcheeta | |
|---|---|---|---|---|---|---|---|---|
| | agent attribute | coordination structure | agent attribute | coordination structure | agent attribute | coordination structure | agent attribute | coordination structure |
| MA-ICQ | 61.3 ± 13.7 | 34.1 ± 18.7 | 83.7 ± 13.1 | 47.8 ± 15.3 | 69.4 ± 14.7 | 41.6 ± 17.3 | 1583.2 ± 547.7 | 712.6 ± 375.2 |
| MA-CQL | 59.4 ± 8.3 | 35.6 ± 14.3 | 62.4 ± 16.7 | 41.2 ± 21.7 | 51.1 ± 11.0 | 37.2 ± 18.5 | 2678.2 ± 900.9 | 948.4 ± 462.7 |
| OMAR | 66.5 ± 9.7 | 38.2 ± 14.9 | 85.6 ± 29.3 | 44.7 ± 26.8 | 71.1 ± 15.2 | 40.9 ± 23.8 | 2295.0 ± 357.2 | 926.3 ± 364.5 |
| MA-SfBC | 69.1 ± 19.2 | 44.8 ± 25.9 | 64.4 ± 21.5 | 40.3 ± 26.1 | 82.0 ± 33.3 | 49.1 ± 29.5 | 2397.4 ± 670.3 | 1083.1 ± 492.5 |
| DOM2 | 74.0 ± 16.1 | 57.2 ± 19.1 | 89.1 ± 21.7 | 64.5 ± 24.2 | 91.8 ± 34.9 | 69.4 ± 26.3 | 3178.7 ± 370.5 | 1748.7 ± 357.4 |
| MADIFF | 66.7 ± 12.9 | 53.9 ± 16.4 | 91.6 ± 18.4 | 65.2 ± 23.9 | 83.1 ± 26.4 | 69.6 ± 28.4 | 2741.9 ± 360.4 | 1427.5 ± 290.3 |
| MCGD | **78.4 ± 14.2** | **65.3 ± 15.6** | **99.3 ± 20.1** | **73.8 ± 19.6** | **104.2 ± 19.4** | **79.5 ± 21.6** | **3518.4 ± 273.0** | **1925.8 ± 318.3** |
| Abs.(%) Avg.↑ | 4.4(5.9) | 8.1(14.2) | 7.7(8.4) | 8.6(13.2) | 12.4(13.5) | 9.9(14.2) | 339.7(10.7) | 177.1(10.1) |

dynamic environmental changes (e.g., agent attributes and coordination structure). For fair evaluation, we run each offline experiment five times with different random seeds and report the average episodic return obtained in online rollout as the performance measure.

### 5.1. Experimental Setup

#### 5.1.1. BENCHMARKS

We conduct extensive evaluations on three following well-established multi-agent benchmarks:

• Multi-Agent Particle Environments (MPE) (Lowe et al., 2017). Multiple 2D particles work together to achieve shared objectives across various scenarios. In the Spread task, three particles are positioned randomly and must cover three landmarks without colliding. In the Tag task, three predators cooperate to capture a pre-trained, faster-moving prey. In the World task, three predators catch a pre-trained prey, which aims to collect food while evading capture. For the offline datasets, we use four datasets from (Pan et al., 2022), each corresponding to different levels of training quality, Expert, Medium-Replay, Medium, and Random.

• Multi-Agent MuJoCo (MAMuJoCo) (Peng et al., 2021). Independent agents control different robotic joints to maximize forward speed. Three configurations are chosen: 2-agent halfcheetah (2halfcheetah), 2-agent ant (2ant), and 4-agent ant (4ant). The offline datasets are sourced from (Formanek et al., 2023), which includes varying quality levels-Good, Medium, and Poor-for each control task.

• StarCraft Multi-Agent Challenge (SMAC) (Samvelyan et al., 2019). A team of agents collaborates to fight against an enemy team controlled by hand-coded AI. Four different maps are explored: three Marines per team (3m), two Stalkers and three Zealots per team (2s3z), five Marines versus six Marines (5m_vs_6m), and eight Marines per team (8m). The off-the-grid offline datasets (Formanek et al., 2023) are used here, with varying quality levels—Good, Medium, and Poor—available for each map.

#### 5.1.2. BASELINE

We compare the MGCD framework with the various state-of-the-art baselines including: offline MARL algorithms such as MA-ICQ (Yang et al., 2021), MA-CQL (Jiang & Lu, 2023), and OMAR (Pan et al., 2022)), the extension of the single-agent diffusion-based policy, MA-SfBC (Chen et al., 2022)), and diffusion-based MARL methods such as DOM2 (Li et al., 2023) and MADIFF (Zhu et al., 2023).

### 5.2. Numerical Results

To evaluate the effectiveness of collaborative policies, we report numerical results for both MCGD and baseline methods, using offline Expert and Good datasets across three environments. The average values and standard deviations of the episodic returns for each model are summarized in Table 1. Notably, our MCGD framework consistently achieves state-of-the-art performance across all multi-agent cooperative scenarios, with smaller deviations in most settings, which demonstrates its advantages in terms of both effectiveness and stability. The fact that the DOM2 and MADIFF algorithms rank second and third in each task underscores the potential of diffusion models in addressing the out-of-distribution challenge in offline multi-agent reinforcement learning. When compared to diffusion-based

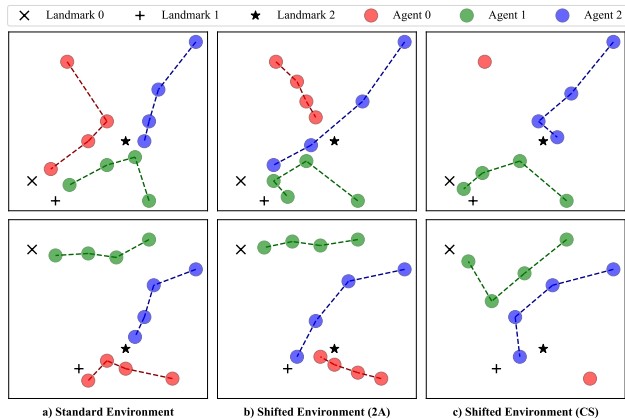

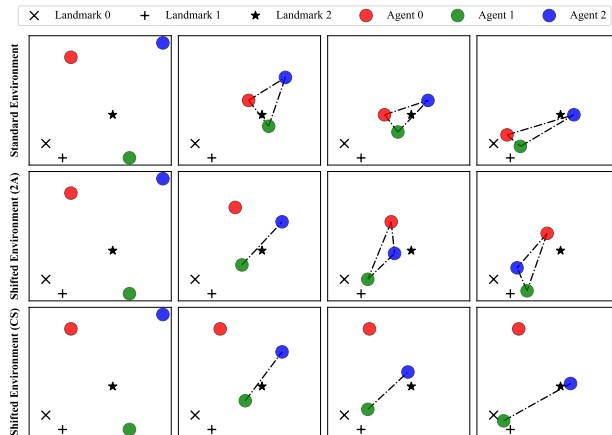

Figure 4: Visualization of two episodic trajectories in the MPE Spread task with three agents and three landmarks, showing: a) the standard environment, b) dynamic agent attributes (2A), and c) dynamic coordination structure (CS) in the shifted environment.

Figure 5: Visualization of dynamic learned coordination graph over timesteps in the MPE Spread task with three agents and three landmarks under different settings: standard environment, dynamic agent attributes, and dynamic coordination structure.

baselines, MCGD achieves improvements of at least 5.7%, 6.5%, and 4.7% in the MPE, MAMujoco, and SMAC benchmarks, respectively. These significant advantages further highlight the importance of the underlying multi-agent coordination framework when applying diffusion models to historical trajectories. The experimental results on Medium and Poor datasets are provided in Appendix.

To further validate the policy robustness, we design shifted environments (details are provided in Appendix 7.3.2) by altering agent attributes and coordination structures. Specifically, we randomly modify the particle speed in the MPE environment and adjust the power, density, and friction of joints in MAMuJoCo to simulate dynamic changes in agent attributes. Additionally, we randomly select a particle or joint to fix its speed or power attribute as zero, simulating changes in coordination structures caused by the sudden disconnection of an agent. For each selected task, we train the collaborative policies using the offline datasets generated in standard environments and evaluate them in shifted environments. The corresponding experimental results are shown in Table 2. We observe that MCGD consistently achieves the highest episodic return across all shifted environments, underscoring the robustness of our framework. Furthermore, in more challenging scenarios with dynamic coordination structures, MCGD's performance advantage becomes even more pronounced. This superiority can be attributed to the modeling of diverse edge structures and node actions within the multi-agent collaboration graph, facilitated by our defined graph diffusion process.

### 5.3. Qualitative Analysis

To investigate the effectiveness and robustness of the MCGD framework, we focus on the MPE spread task involving 3 agents and 3 landmarks, conducted in both standard and shifted environments. The team reward is defined as the sum of the distances between each landmark and its closest agent. When an agent collides with another, it incurs a penalty, which is subtracted from the team reward. Two example episodic trajectories are visualized in Figure 4, illustrating the coordinative behavior of agents in the task.

In the standard environment, each agent quickly approaches one or two landmarks to minimize the distance to the closest landmark. When the minimum speed of Agent 0 decreases, anisotropic diffusion within the MCGD framework dynamically reallocates the target landmarks for all three agents, altering their motion trajectories. Additionally, the diffusion process's neighbor dependence helps prevent collisions caused by excessive action uncertainty when two agents are close to one another. This cooperation ensures the successful completion of the task.

When Agent 0 goes offline and can no longer move, categorical diffusion alters the coordination structure among the remaining agents. With Agent 0's influence removed, the remaining two agents shift their focus from minimizing the distance to a single landmark to minimizing the sum of distances to multiple landmarks.

To illustrate how the learned coordination graph evolves during task execution, we further provide a case study on the MPE Spread task in Figure 5, where the horizontal axis indicates timesteps and the vertical axis shows different experimental settings.

Initially, although a nearest-neighbor graph is used for initialization, large positional differences between agents cause the forward diffusion process to disrupt coordination edges. As a result, the model predicte no edges, and agents act independently. As agents move closer, the learned graph

Table 3: Comparison of different processing methods for neighboring observations—average observation (AO) and feature concatenation (FC)—on offline Good datasets in the SMAC benchmark, in terms of average reward and computation time.

| SMAC | 3m | | 2s3z | | 5m6m | | 8m | |
|---|---|---|---|---|---|---|---|---|
| | average reward | spent time | average reward | spent time | average reward | spent time | average reward | spent time |
| MCGD-AO | $22.1 \pm 0.1$ | $174.38 \pm 10.45$ | $20.7 \pm 0.1$ | $179.01 \pm 9.83$ | $18.9 \pm 0.5$ | $177.52 \pm 14.77$ | $20.1 \pm 0.2$ | $184.74 \pm 11.37$ |
| MCGD-FC | $21.3 \pm 0.3$ | $195.40 \pm 8.77$ | $20.5 \pm 0.1$ | $203.18 \pm 16.04$ | $18.4 \pm 0.4$ | $207.54 \pm 13.80$ | $19.4 \pm 0.4$ | $210.39 \pm 10.08$ |

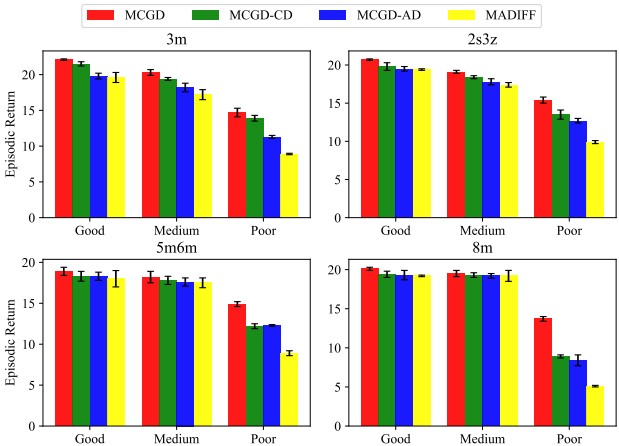

Figure 6: Ablation study on categorical diffusion and anisotropic diffusion within the MCGD framework, evaluated in SMAC environments.

gradually recover the underlying coordination structure, enabling effective collaboration such as landmark assignment and collision avoidance.

In a modified setting where the speed of Agent 0 is reduced, coordination edges emerge primarily between the other agents. Agent 0, due to its reduced speed, require more timesteps to engage in coordination, delaying the full coordination graph reconstruction. In a more extreme case where Agent 0 is inactive, it remains isolated, and coordination is exclusively formed between the other two agents. These visualizations demonstrate the adaptive nature of the learned graph, which dynamically reflects the agents' interaction context and task demands.

### 5.4. Ablation Studies

To verify the functionality of key modules, categorical diffusion and anisotropic diffusion, we have designed two model variants, MCGD-CD and MCGD-AD. Specifically, in MCGD-CD, we ignore the dynamics of the multi-agent coordination structure and fix it to the k-nearest neighboring graph, whereas in MCGD-AD, we perform independent diffusion on multi-agent trajectories and aggregate the observations from neighboring agents based on the coordination structure. We experimentally compare the MCGD framework and its variants with the best performing baseline, MADIFF, using offline datasets categorized as Good, Medium, and Poor in the SMAC environment. As shown in Figure 6, the full MCGD framework outperforms all datasets, with both variants also surpassing the baseline,

highlighting the importance of the two key modules in the MCGD framework. Although MCGD-CD performs satisfactorily in simpler tasks such as 3m and 2s3z, it struggles in other complex scenarios, reflecting the limitations of fixed coordination structures in multi-agent collaboration and highlighting the need for modeling coordination structure diversity in MCGD.

We further validate the design of processing neighboring observations through an ablation study using the SMAC benchmark, comparing two variants: MCGD-AO (with observation averaging) and MCGD-FC (with feature concatenation). As shown in Table 3, MCGD-AO achieves both higher average rewards and lower computational cost across all tasks. The inferior performance of MCGD-FC is largely attributed to increased parameterization and greater optimization difficulty, further supporting the effectiveness and generality of our averaging-based approach.

## 6. Conclusion

This work proposes the first graph diffusion model for offline multi-agent coordination, which employs categorical diffusion on discrete edge attributes and anisotropic diffusion on continuous node attributes to capture structure diversity and action diversity, respectively. Extensive evaluations across three benchmark multi-agent environments demonstrate the effectiveness of the MCGD framework in coordination and its superior robustness to dynamic changes in environmental conditions.

In future work, we aim to introduce more complex and adaptable collaboration graph structures to support a broader range of agent interactions. Real-world validation remains a critical direction to further demonstrate the practical applicability of our framework. Our team is currently working on deploying the proposed method in real-world multi-robot hunting scenarios. Although quantitative results are not yet available for inclusion in this version, we are actively collecting data and refining the deployment process. We plan to report these findings as part of a more extensive evaluation in a future extension of this work.

## Impact Statement

This paper contributes to advancing the field of Machine Learning. While there are numerous potential societal implications of our research, we believe none require specific emphasis in this context.

## Acknowledgments

This work was supported by NSFC Projects (Nos. 92248303, 92370124, 62350080, 62441612, 62322202), and 62432006, BNRist (BNR2022RC01006) and National Key Laboratory under grant 241-HF-D07-01.

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

# 7. Appendix

## 7.1. Framework Detail

### 7.1.1. NOTATIONS

To facilitate comprehension, we summarize the primary notations related to multi-agent reinforcement learning and graph diffusion models, along with their corresponding descriptions, in Table 4.

Table 4: Glossary of Notations.

| Notation | Description |
|----------|-------------|
| $\mathcal{M}$ | Markov decision process |
| $n; \mathcal{I}$ | Single agent; Agent set |
| $\mathcal{S}; \mathcal{A}$ | State and action spaces |
| $\mathcal{P}; \mathcal{O}$ | Transition and observation functions |
| $\mathcal{R}; \gamma$ | Reward function; Discount factor |
| $O; S; A$ | Multi-agent observations, states, and actions |
| $o, s, a$ | Single-agent observation, state, and action |
| $\mathcal{D}; \tau$ | Replay buffer; Historical trajectory |
| $X$ | Random variable |
| $q; p$ | Prior and posterior distributions |
| $\mathcal{N}; Q$ | Gaussian noise; Transition matrix |
| $\mu; \Sigma$ | Average value and covariance matrix |
| $G; E$ | Coordination graph; Coordination edges |
| $C; D$ | Similarity matrix; Diagonal degree matrix |
| $K, T$ | Diffusion iterations; Timesteps |
| $f$ | Graph denoising network |
| $\mathcal{L}$ | Loss function |

### 7.1.2. MCGD TRAINING PROCESS

We summarize the training process of MCGD framework in Algorithm 2. At each training episode, we randomly sample a multi-agent historical trajectory $\tau$ from the offline dataset $\mathcal{D}$. Within this trajectory $\tau$, we initialize a nearest-neighboring coordination graph $G_t = (A_t, E_t)$ at each timestep $t$ and sample a parameter $K$ as the number of diffusion iterations (lines 5 and 6 in Algorithm 2). For the discrete edges $E_t$, we apply the closed-form expression of categorical diffusion to calculate the perturbed structure $E_K^t$ (line 7 in Algorithm 2). For each agent $n_i \in \mathcal{I}$, we calculate the covariance matrix $\Sigma_i$, which defines the anisotropic structure of the Gaussian noise applied to its action space (line 9 in Algorithm 2). We then employ the closed-form continuous diffusion on the agent's action $a_i^t$ to obtain the noisy action $a_K^t$ (line 10 in Algorithm 2). Integrating the perturbed edges and each agent's action, we derive the perturbed coordination graph $G_K^t = (A_K^t, E_K^t)$ and optimize the graph diffusion model by minimizing the training losses $\mathcal{L}_{ce}$ and $\mathcal{L}_{ad}$ (lines 13 and 14 in Algorithm 2).

---

**Algorithm 2** MCGD Training Algorithm

---

1: **Input:** offline multi-agent dataset $\mathcal{D}$
2: **Initialize:** graph diffusion model $f_\theta$ and Q-value function $\mathcal{Q}_{\phi_i}$ for each agent $n_i$
3: **for** each training episode **do**
4:     **for** each training step $t$ **do**
5:         Construct the nearest coordination graph $G_t$
6:         Sample the diffusion parameters $K$
7:         Calculate the perturbed edge attribute $E_K^t$
8:         **for** each agent $n_i$ **do**
9:             Calculate the covariance matrix $\Sigma_i$
10:            Calculate the noisy single-agent action $a_{i,K}^t$
11:            Optimize the Q value function $\mathcal{Q}_{\phi_i}$
12:         **end for**
13:         Derive the perturbed coordination graph $G_K^t$
14:         Optimize the graph diffusion model $f_\theta$ by minimizing $\mathcal{L}_{ce}$ and $\mathcal{L}_{ad}$
15:     **end for**
16: **end for**

---

## 7.2. Theorem Proof

### 7.2.1. PROOF FOR THEOREM 4.1

*Proof.* We begin by considering the Taylor expansion of the matrix exponential for $Q_t^E$:

$$Q_t^E = e^{(C-D)t} = I + \sum_{i=1}^{\infty} \left[ \frac{(C-D)^i t^i}{i!} \right], \qquad (17)$$

where $I$ denotes the identity matrix.

**Symmetry.** Since both the similarity matrix $C$ and the diagonal degree matrix $D$ are symmetric, it follows that:

$$(C-D)^T = (C-D). \qquad (18)$$

Thus, the transpose of $Q_t^E$ is:

$$\begin{aligned} \left[ Q_t^E \right]^T &= I^T + \sum_{i=1}^{k} \left[ \frac{t^i}{i!} \cdot \left[ (C-D)^i \right]^T \right] \\ &= I + \sum_{i=1}^{k} \left[ \frac{t^i}{i!} \cdot (C-D)^i \right] \\ &= Q_t^E. \end{aligned} \qquad (19)$$

Therefore, $Q_t^E$ is symmetric.

**Additivity.** Using the properties of the matrix exponential, we have:

$$\begin{aligned} Q_{t_i+t_j}^E &= e^{(C-D)(t_i+t_j)} \\ &= e^{(C-D)t_i} e^{(C-D)t_j} \\ &= Q_{t_i}^E Q_{t_j}^E. \end{aligned} \qquad (20)$$

Thus, $Q_t^E$ satisfies the additivity property.

**Locality.** From the Taylor expansion, each term in the series for $Q_t^E$ involves powers of $t$. For $i > 1$, we have:

$$\lim_{t \to 0} \frac{(C-D)^i t^i}{i!} = \frac{(C-D)^i 0^i}{i!} = 0. \qquad (21)$$

Therefore, as $t \to 0$, the transition matrix reduces to:

$$\lim_{t \to 0} Q_t^E = \lim_{t \to 0} \left[ I + \sum_{i=1}^{\infty} \left[ \frac{(C-D)^i t^i}{i!} \right] \right] = I. \qquad (22)$$

This establishes the locality of $Q_t^E$ as $t$ approaches zero.

**Convergence.** First, note that the matrix $C - D$ has an eigenvalue of $0$:

$$(C-D)\mathbf{1} = C\mathbf{1} - D\mathbf{1} = \mathbf{0} = 0 \cdot \mathbf{1}, \qquad (23)$$

where $\mathbf{1}$ and $\mathbf{0}$ are the vectors of all ones and zeros, respectively. If $C - D$ is irreducible, then the geometric multiplicity of the eigenvalue $0$ is $1$, and the corresponding normalized eigenvector is $\frac{1}{|\mathcal{I}|}\mathbf{1}$.

Next, we show that $C - D$ is a semi-negative definite. For any vector $x \in \mathbb{R}^{|\mathcal{I}|}$, we have:

$$\begin{aligned} x^T(C-D)x &= x^T C x - x^T D x \\ &= \sum_{i,j=1}^{|\mathcal{I}|} C_{i,j} x_i x_j - \sum_{i=1}^{|\mathcal{I}|} D_{i,i} x_i^2 \\ &= -\frac{1}{2} \left[ \sum_{i=1}^{|\mathcal{I}|} D_{i,i} x_i^2 - \sum_{i,j=1}^{|\mathcal{I}|} 2 C_{i,j} x_i x_j + \sum_{j=1}^{|\mathcal{I}|} D_{j,j} x_j^2 \right] \\ &= -\frac{1}{2} \left[ \sum_{i,j=1}^{|\mathcal{I}|} \left( C_{i,j} \left( x_i - x_j \right)^2 \right) \right] \\ &\leq 0. \end{aligned} \qquad (24)$$

Thus, $C - D$ is semi-negative definite, and all eigenvalues except for the one corresponding to $1$ are negative.

We can now express $Q_t^E$ in terms of the eigenvalue decomposition of $C - D$:

$$Q_t^E = e^{(C-D)t} = V e^{\Lambda t} V^T, \qquad (25)$$

where $V$ is the matrix of eigenvectors of $C - D$ and $\Lambda$ is the diagonal matrix of eigenvalues $0 = \lambda_1 > \lambda_2 \geq \cdots \geq \lambda_{|\mathcal{I}|}$.

As $t \to \infty$, the matrix exponential $e^{\Lambda t}$ behaves as follows:

$$\lim_{t \to \infty} e^{\lambda_i t} = \begin{cases} 1 & \text{if } i = 1, \\ 0 & \text{if } i > 1. \end{cases} \qquad (26)$$

Therefore, in the limit as $t \to \infty$, the matrix $Q_t^E$ converges to the rank-1 matrix formed by the eigenvector corresponding to the zero eigenvalue:

$$\lim_{t \to \infty} Q_t^E = \frac{1}{\mathcal{I}} \mathbf{1}\mathbf{1}^T. \qquad (27)$$

This completes the proof of the convergence of $Q_t^E$ as $t \to \infty$ when $C - D$ is irreducible. $\square$

### 7.2.2. FORWARD ANISOTROPIC PROCESS

*Proof.* Given the anisotropic noising process at iteration $k$, we derive the following expression for $a_{i,k}^t$:

$$a_{i,k}^t = \sqrt{1 - \beta_k} a_{i,k-1}^t + \sqrt{\beta_k} \epsilon_{i,k}, \qquad (28)$$

where $\epsilon_{i,k} \sim \mathcal{N}(0, \Sigma_i)$ represents the anisotropic noise of agent $n_i$.

By recursively unrolling the forward process, we rewrite $a_{i,k}^t$ as follows:

$$
\begin{aligned}
a_{i,k}^t &= \sqrt{1 - \beta_k} a_{i,k-1}^t + \sqrt{\beta_k} \epsilon_{i,k} \\
&= \prod_{j=1}^{k} \sqrt{1 - \beta_j} a_{i,0}^t + \sum_{j=1}^{k} \sqrt{\beta_j (1 - \beta_k) \cdots (1 - \beta_1)} \epsilon_{i,j}.
\end{aligned}
\qquad (29)
$$

Using $\alpha_k = 1 - \beta_k$ and $\overline{\alpha}_k = \prod_{i=1}^{k} \alpha_i$, we express the marginal distribution of $a_{i,k}^t$ as:

$$a_{i,k}^t = \sqrt{\overline{\alpha}_k} a_{i,0}^t + \sum_{j=1}^{k} \sqrt{\beta_j \frac{\overline{\alpha}_k}{\alpha_{i,j}}} \epsilon_j. \qquad (30)$$

Since each noise term $\epsilon_{i,j}$ is independently sampled from $\mathcal{N}(0, \Sigma_i)$, we know that:

$$\mathbb{E}[\epsilon_{i,j}] = 0, \quad \mathrm{Cov}(\epsilon_{i,j}) = \Sigma_i. \qquad (31)$$

The mean and variance of the weighted sum of independent and identically distributed Gaussian noises are given by:

$$\mathbb{E}\left[ \sum_{j=1}^{k} \sqrt{\beta_j \frac{\overline{\alpha}_k}{\overline{\alpha}_j}} \epsilon_{i,j} \right] = \sum_{j=1}^{k} \sum_{j=1}^{k} \sqrt{\beta_j \frac{\overline{\alpha}_k}{\overline{\alpha}_j}} \mathbb{E}[\epsilon_{i,j}] = 0, \qquad (32)$$

$$
\begin{aligned}
\mathrm{Cov}\left( \sum_{j=1}^{k} \sqrt{\beta_j \frac{\overline{\alpha}_k}{\overline{\alpha}_j}} \epsilon_{i,j} \right) &= \sum_{j=1}^{k} \beta_j \frac{\overline{\alpha}_k}{\overline{\alpha}_j} \mathrm{Cov}(\epsilon_{i,j}) \\
&= \sum_{j=1}^{k} \beta_j \frac{\overline{\alpha}_k}{\overline{\alpha}_j} \Sigma_i \\
&= \sum_{j=1}^{k} \alpha_k \cdots \alpha_{j+1} (1 - \alpha_j) \Sigma_i \\
&= (1 - \overline{\alpha}_k) \Sigma_i.
\end{aligned}
\qquad (33)
$$

Therefore, the distribution of this weighted sum of Gaussian noises is given by:

$$\sum_{j=1}^{k} \sqrt{\beta_j \frac{\overline{\alpha}_k}{\overline{\alpha}_j}} \epsilon_{i,j} \sim \mathcal{N}(0, (1 - \overline{\alpha}_k) \Sigma_i), \qquad (34)$$

Finally, based on the previous steps, the marginal distribution of $a_{i,k}^t$ can be written as follows:

$$q(a_{i,k}^t | a_{i,0}^t) = \mathcal{N}(a_{i,k}^t; \sqrt{\overline{\alpha}_k} a_{i,0}^t, (1 - \overline{\alpha}_k) \Sigma_i). \qquad (35)$$

$\square$

### 7.3. Environmental Details

#### 7.3.1. IMPLEMENTATION DETAILS

All experiments are conducted on Linux servers with a 64-core Intel Xeon Platinum 8336C CPU (2.30 GHz) and an NVIDIA A800-SXM4-80GB GPU. Across all experiments, the diffusion parameter $K$ is varied within the range $[50, 200]$, with a learning rate of $2e-4$, a batch size of 32, a reward discount $\gamma$ of 0.99, and the Adam optimizer utilized.

For the MPE experiments, we employ the earlier version of the environment provided by OMAR (Pan et al., 2022), where agents receive distinct, environment-specific rewards that depend on their individual actions and contributions to the collective task. To ensure fairness and consistency in comparisons, we utilize OMAR's datasets and environments, ensuring all baseline models were trained and evaluated under identical conditions.

In the MAMuJoCo experiments, we rely on an off-the-shelf dataset (Formanek et al., 2023), which includes Good, Medium, and Poor quality datasets for each task. Each dataset is generated using three independently trained MA-TD3 policies (Ackermann et al., 2019), supplemented with exploration noise to enhance behavioral diversity.

Similarly, for the SMAC experiments, we utilize a separate off-the-shelf dataset (Formanek et al., 2023) that includes Good, Medium, and Poor quality datasets for each map. These datasets are constructed using three independently trained QMIX policies (Rashid et al., 2020), with exploration noise deliberately introduced to the policies to encourage behavioral diversity.

#### 7.3.2. SHIFTED ENVIRONMENTS

The parameter settings for agent attributes are summarized in Table 5. In the MPE experiments, we design the shifted Spread, Tag, and World environments by reducing the minimum speed of a single random agent from the original value of 0.8 to 0.3. For the MAMuJoCo experiments, we design the shifted 2halfcheetah environment following prior work (Packer et al., 2018). Specifically, we randomly sample values for the following three parameters: power (influencing the multiplied force), density (affecting the weight), and friction (determining the sliding friction of the joints).

Table 5: Parameter setting in the shifted MPE and MAMuJoCo environments.

| **MPE** | **Standard** | **Shifted** |
|---|---|---|
| Speed | $[0.8, 1.0]$ | $[0.3, 1.0]$ |
| **MAMuJoCo** | **Standard** | **Shifted** |
| Power | 1.0 | $[0.8, 1.2]$ |
| Density | 1000 | $[750, 1250]$ |
| Friction | 0.4 | $[0.25, 0.55]$ |

## 7.4. Supplemental Experiments

### 7.4.1. SENSITIVE ANALYSIS

To investigate the effect of the regularization coefficient $\lambda$ on our framework's performance, we increase its value from 0.1 to 20.0 in the MPE Tag task, evaluating performance at the Random, Medium-Replay, Medium, and Expert levels. We then report the resulting episodic returns to evaluate performance at each setting, as shown in Table 7. For the expert dataset, smaller values of $\lambda$ lead to better performance, because the policy is already well-trained, and the regularization does not need to be as strong. In contrast, larger values of $\lambda$ tend to improve performance in the other datasets, as regularization plays a more significant role in preventing overfitting and stabilizing offline learning.

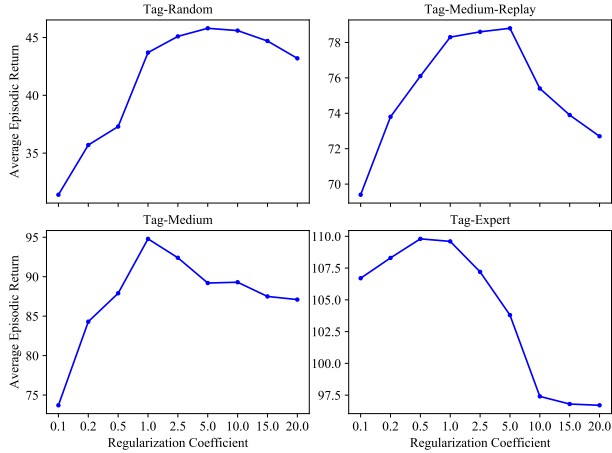

Figure 7: Sensitivity analysis on the $\lambda$ value in offline datasets from the Tag task at the Random, Medium-Replay, Medium, and Expert levels.

### 7.4.2. SAMPLING EFFICIENCY

To assess the efficiency and scalability of our framework, we incrementally increase the number of agents and record the time required (in ms) for the policy sampling process in both our graph diffusion model and the baseline diffusion model, MADIFF, as shown in Table 6. As shown in the table, while our proposed graph-based diffusion model introduces some additional time overhead compared to the baseline, the gap remains relatively small and stable as the number of agents increases. This demonstrates the sampling efficiency and scalability of the MCGD framework.

Table 6: Wall-clock time spent on sampling multi-agent policies in the MCGD and MADIFF models.

| Method | 8 | 16 | 32 | 64 |
|---|---|---|---|---|
| MADIFF | 145.47 | 149.36 | 148.19 | 149.84 |
| MCGD | 153.69 | 154.27 | 155.38 | 154.94 |

Table 7: Comparison between MCGD and baselines on offline Medium-Replay, Medium, and Random datasets across the MPE benchmark: "average value $\pm$ standard deviation". **Bold**: the best performance, underline: the second performance.

| Method | Random | | | Medium-Replay | | | Medium | | |
|---|---|---|---|---|---|---|---|---|---|
| | Spread | Tag | World | Spread | Tag | World | Spread | Tag | World |
| MA-ICQ | $6.3 \pm 3.5$ | $2.2 \pm 2.6$ | $1.0 \pm 3.2$ | $11.2 \pm 8.4$ | $28.3 \pm 19.2$ | $9.7 \pm 6.4$ | $24.5 \pm 8.6$ | $55.8 \pm 19.3$ | $53.7 \pm 21.8$ |
| MA-CQL | $24.0 \pm 9.8$ | $5.0 \pm 8.2$ | $0.6 \pm 2.0$ | $15.3 \pm 7.6$ | $22.8 \pm 19.4$ | $15.9 \pm 14.2$ | $27.9 \pm 8.1$ | $57.4 \pm 21.8$ | $44.3 \pm 14.1$ |
| OMAR | $34.4 \pm 5.3$ | $11.1 \pm 2.8$ | $5.9 \pm 5.2$ | $35.6 \pm 15.2$ | $41.6 \pm 17.9$ | $21.1 \pm 15.6$ | $41.7 \pm 21.3$ | $53.4 \pm 25.9$ | $45.6 \pm 16.0$ |
| MA-SfBC | $5.1 \pm 3.9$ | $11.6 \pm 5.1$ | $7.4 \pm 3.9$ | $8.2 \pm 4.6$ | $12.7 \pm 7.3$ | $9.1 \pm 5.9$ | $51.6 \pm 14.2$ | $47.1 \pm 17.9$ | $54.2 \pm 22.7$ |
| DOM2 | $\underline{37.4} \pm 11.3$ | $\underline{29.6} \pm 8.1$ | $\underline{38.9} \pm 11.8$ | $\underline{63.1} \pm 9.5$ | $\underline{68.2} \pm 16.7$ | $\underline{65.9} \pm 10.6$ | $\underline{78.6} \pm 8.1$ | $\underline{82.6} \pm 18.2$ | $84.5 \pm 23.4$ |
| MADIFF | $7.2 \pm 3.6$ | $4.6 \pm 2.6$ | $0.7 \pm 3.1$ | $35.1 \pm 7.2$ | $53.9 \pm 11.4$ | $56.4 \pm 12.5$ | $60.3 \pm 10.6$ | $72.7 \pm 9.4$ | $\underline{87.2} \pm 13.9$ |
| MCGD | $\mathbf{41.8} \pm 11.7$ | $\mathbf{43.7} \pm 11.8$ | $\mathbf{48.3} \pm 10.1$ | $\mathbf{74.4} \pm 10.3$ | $\mathbf{78.3} \pm 9.6$ | $\mathbf{79.3} \pm 14.2$ | $\mathbf{88.6} \pm 9.3$ | $\mathbf{94.8} \pm 12.4$ | $\mathbf{99.4} \pm 12.8$ |

Table 8: Comparison between MCGD and baselines on offline Poor and Medium datasets across the MAMuJoCo benchmark: "average value $\pm$ standard deviation". **Bold**: the best performance, underline: the second performance.

| Method | Poor | | | Medium | | |
|---|---|---|---|---|---|---|
| | 2**halfcheetah** | 2**ant** | 4**ant** | 2**halfcheetah** | 2**ant** | 4**ant** |
| MA-ICQ | $271.4 \pm 183.6$ | $583.4 \pm 327.9$ | $658.4 \pm 392.7$ | $749.2 \pm 296.4$ | $581.4 \pm 257.4$ | $1025.8 \pm 437.1$ |
| MA-CQL | $293.5 \pm 138.6$ | $437.6 \pm 291.4$ | $593.1 \pm 385.5$ | $963.4 \pm 316.6$ | $638.2 \pm 244.9$ | $900.4 \pm 281.6$ |
| OMAR | $362.7 \pm 314.2$ | $837.5 \pm 194.2$ | $509.4 \pm 241.4$ | $2797.0 \pm 445.7$ | $772.5 \pm 216.4$ | $917.3 \pm 349.2$ |
| MA-SfBC | $216.4 \pm 208.5$ | $883.1 \pm 372.5$ | $976.4 \pm 241.3$ | $1386.8 \pm 248.8$ | $1038.4 \pm 294.5$ | $1529.4 \pm 371.6$ |
| DOM2 | $803.0 \pm 274.6$ | $916.2 \pm 181.6$ | $1157.9 \pm 224.5$ | $\underline{2851.2} \pm 145.5$ | $\underline{1431.7} \pm 304.8$ | $1691.5 \pm 183.3$ |
| MADIFF | $\underline{814.2} \pm 245.7$ | $\underline{934.8} \pm 174.2$ | $\underline{1362.8} \pm 274.3$ | $2194.7 \pm 319.6$ | $1247.3 \pm 358.1$ | $\underline{1728.9} \pm 249.3$ |
| MCGD | $\mathbf{1244.7} \pm 159.2$ | $\mathbf{1479.3} \pm 164.9$ | $\mathbf{1738.4} \pm 149.2$ | $\mathbf{3168.4} \pm 210.8$ | $\mathbf{1867.2} \pm 247.1$ | $\mathbf{2074.3} \pm 156.2$ |

### 7.4.3. Additional Results

For a comprehensive evaluation of MCGD and baselines, we conduct comparative experiments across offline datasets from various difficulty levels in the MPE, MAMuJoCo, and SMAC benchmarks. Specifically, we report collaborative performance on the Random, Medium-Replay, and Medium datasets in the Spread, Tag, and World tasks from the MPE benchmark (Table 7), performance on the Poor and Medium datasets in the 2halfcheetah, 2ant, and 4ant tasks from the MAMuJoCo benchmark (Table 8), and performance on the Poor and Medium datasets in the 3m, 2s3z, 5m6m, and 8m tasks from the SMAC benchmark (Table 9). As shown in these tables, our MCGD framework consistently and significantly outperforms all baseline methods across different datasets and environments, improving the average episodic return by at least 4.1, 317.2, and 0.3 in MPE, MAMuJoCo, and SMAC, respectively. In the most challenging (Random or Poor) scenarios, MCGD increases the average returns in MPE from 35.3 to 44.6 in MPE, in MAMuJoCo from 1037.3 to 1487.5, and in SMAC from 9.1 to 10.3, achieving average improvements of 9.3(26.3%), 450.2(43.4%), and 1.2(13.2%) over baselines and demonstrating the effectiveness advantage of our method.

Table 9: Comparison between MCGD and baselines on offline Poor and Medium datasets across the SMAC benchmark: "average value $\pm$ standard deviation". **Bold**: the best performance, underline: the second performance.

| Method | Poor | | | | Medium | | | |
|---|---|---|---|---|---|---|---|---|
| | **3m** | **2s3z** | **5m6m** | **8m** | **3m** | **2s3z** | **5m6m** | **8m** |
| MA-ICQ | $4.9 \pm 0.5$ | $7.8 \pm 0.3$ | $9.1 \pm 0.6$ | $6.4 \pm 0.3$ | $18.1 \pm 0.7$ | $17.2 \pm 0.6$ | $15.3 \pm 0.7$ | $18.6 \pm 0.8$ |
| MA-CQL | $5.8 \pm 0.4$ | $\underline{10.1} \pm 0.7$ | $\underline{10.4} \pm 1.0$ | $4.6 \pm 2.4$ | $\underline{18.9} \pm 0.7$ | $14.3 \pm 2.0$ | $17.0 \pm 1.2$ | $16.8 \pm 3.1$ |
| OMAR | $4.7 \pm 0.4$ | $5.8 \pm 0.6$ | $9.6 \pm 1.2$ | $5.9 \pm 0.5$ | $16.3 \pm 0.5$ | $14.5 \pm 0.4$ | $14.8 \pm 0.6$ | $16.3 \pm 0.7$ |
| MA-SfBC | $5.2 \pm 0.4$ | $7.4 \pm 0.6$ | $8.1 \pm 0.3$ | $\underline{6.9} \pm 0.7$ | $17.4 \pm 0.9$ | $16.8 \pm 0.5$ | $16.5 \pm 0.5$ | $18.1 \pm 0.4$ |
| DOM2 | $\underline{9.1} \pm 0.2$ | $9.6 \pm 0.3$ | $8.9 \pm 0.2$ | $5.7 \pm 0.4$ | $18.7 \pm 0.4$ | $17.1 \pm 0.5$ | $17.2 \pm 0.8$ | $18.0 \pm 0.7$ |
| MADIFF | $8.9 \pm 0.1$ | $9.9 \pm 0.2$ | $8.9 \pm 0.3$ | $5.1 \pm 0.1$ | $17.2 \pm 0.7$ | $\underline{17.4} \pm 0.3$ | $\underline{17.5} \pm 0.6$ | $\underline{19.2} \pm 0.7$ |
| MCGD | $\mathbf{10.4} \pm 0.2$ | $\mathbf{11.3} \pm 0.5$ | $\mathbf{10.7} \pm 0.8$ | $\mathbf{8.8} \pm 0.3$ | $\mathbf{20.6} \pm 0.5$ | $\mathbf{18.8} \pm 0.7$ | $\mathbf{18.3} \pm 0.6$ | $\mathbf{19.7} \pm 0.4$ |

