# OpenReview forum: "Graph Diffusion for Robust Multi-Agent Coordination"
_ICML.cc/2025/Conference — ICML 2025 spotlightposter_

### Official Review · Reviewer_DRJr · 2025-03-12

**Overall Recommendation:** 4

**Summary:**

This paper introduces MCGD (Multi-agent Coordination based on Graph Diffusion), which is a novel framework for offline multi-agent reinforcement learning (MARL) that aims to improve coordination effectiveness and robustness of the policies in dynamic environments. Specifically, MCGD uses graph to model the relationship and coordination of agents. When doing sampling, MCGD uses categorical diffusion to model discrete edge attributes (the correlation between agents) and use anisotropic diffusion to model continuous node attribute (actions of the agents). The authors conducted extensive experiments in different platforms to demonstrate the outstanding performance compared with benchmarks.

## Score updated after Rebuttal

**Claims And Evidence:**

The authors claim that by using MCGD the coordination of agents can be improved especially in dynamic environments. To show this, in the introduction part of the paper, the authors mentioned the case where one agent suddenly becomes unavailable to demonstrate the importance of designing algorithms to tackle general dynamic environments. However, such environments are not tested in the experiments according to the shifted environments in appendix 7.4.2. It would be nice if the authors could have more experiments about changing the number of agents (agents become unavailable or adding new agents to the env) to demonstrate the capability of dealing with environmental changes.

Besides, the agent number of the experiments is greatly limited. In MPE envs, the number of agents is less than 10. Although the authors listed the sampling time comparison result when the number of agents varies from 8 to 64 in 7.5.2, but the planning performance of these experiments is not discussed. In order to better demonstrate the generalizability of the framework, experiments and planning performance comparison when the number of agents is different is beneficial.

**Essential References Not Discussed:**

References look good to me.

**Experimental Designs Or Analyses:**

See Claims And Evidence part of the review.

**Methods And Evaluation Criteria:**

The authors tested the proposed framework in multiple simulation platforms with benchmarks in different tasks, which is pretty comprehensive and clear. It would be better if more experiments, especially real world robotics experiments, can be included.

**Other Comments Or Suggestions:**

See above.

**Other Strengths And Weaknesses:**

The paper is written and organized clearly with useful illustrations and diagrams.

**Questions For Authors:**

See above

**Relation To Broader Scientific Literature:**

This paper studies the MARL problem, which is an important method in multi-agent motion planning. The proposed MCGD framework improves the coordination performance and robustness in dynamic environment, which is important for future real world applications and generalization to more practical scenarios.

**Theoretical Claims:**

The proofs look good to me.

---

> ### Author Rebuttal · Authors · 2025-03-31
>
> We sincerely thank the reviewer for the constructive feedback. We have addressed all comments and revised the manuscript accordingly. Responses are organized by reviewer Weaknesses (W) and Questions (Q), with relevant figures and tables provided in the [anonymized supplementary material](https://anonymous.4open.science/r/MCGD_Rebuttul-23DF).
>
> $\bullet$ W1: Testing Environments with Dynamic Agent Numbers.
> We would like to clarify that our original manuscript already includes evaluation in dynamic settings where agent availability changes.
> Specifically, during testing on the MPE Spread task, we randomly selected one of the three agents and set its velocity to zero to simulate a sudden offline event.
> The corresponding results are reported in the “Coordination Structure” column of Table 2.
> We believe the confusion may stem from Appendix 7.4.2, which focuses on attribute variation (e.g., speed changes) rather than agent removal.
>
> To further address your concern, we have extended our evaluation by increasing the number of agents and landmarks to 8.
> During testing, we randomly deactivate 1 to 4 agents by setting their velocities to zero.
> As shown in Table 4 of the anonymized supplementary material, MCGD consistently outperforms baselines such as DOM2 and MADIFF under all conditions.
> Notably, the performance gap widens as more agents go offline, highlighting MCGD’s robustness and adaptability in highly dynamic environments.
>
> These results support our claim that MCGD is well-suited for handling general coordination under dynamic agent configurations.
>
> $\bullet$ W2: Scalability Comparison.
> While our core experiments follow prior diffusion-based offline MARL settings [1-3], which typically involve fewer than 10 agents, this does not indicate a limitation of our framework in large-scale scenarios.
>
> In Appendix 7.4.2, we first evaluate computational efficiency by comparing sampling time under increasing agent numbers (8 to 64).
> Despite the added complexity of structural diffusion, MCGD remains competitive with existing baselines such as MADIFF.
>
> To further assess planning performance at scale, we additionally conduct experiments on the MPE Spread task with 8, 16, 32, and 64 agents.
> As reported in Table 1 of the anonymized supplementary material, MCGD consistently outperforms MADIFF and DOM2 across all settings, with the performance gap widening as the number of agents increases.
> This trend highlights MCGD’s ability to effectively model complex collaboration patterns under growing agent populations.
>
> These results confirm that our framework is not only computationally scalable, but also capable of maintaining strong coordination performance in large-scale environments.
>
> $\bullet$ W3: Real World Applications:
> We agree that real-world validation is an important direction to further demonstrate the practical applicability of our framework.
>
> Our team is currently working on deploying the proposed method in real-world multi-robot hunting scenarios.
> While we do not yet have quantitative results ready for inclusion in this version, we are actively collecting data and refining the deployment process.
> We plan to report these findings as part of a more extensive evaluation in a future journal extension of this work.
>
> References:
>
> [1] Beyond Conservatism: Diffusion Policies in Offline Multi-agent Reinforcement Learning, Li et al, CoRR 2023.
>
> [2] Madiff: Offline multi-agent learning with diffusion models, Zhu et al, NeurIPS 2024.
>
> [3] Diffusion-based Episodes Augmentation for Offline Multi-Agent Reinforcement Learning, Oh et al, ICML 2024.

---

> > ### Comment · Reviewer_DRJr · 2025-04-01
> >
> > Thank you for the authors' responses. The provided figures and tables look good to me and address my concerns.
> >
> > I have one remaining follow-up question about W1. About the experiments with dynamic agent number, will "add more agents" be different? Indeed, setting and fixing speed to zero can simulate the case where some agents become offline in execution, but in many long-horizon tasks, especially in life-long execution cases, adding more agents will also be worth studying and interesting. Will this case be theoretically and empirically different than the existing experiments? If yes, could you analyze this case? If not, could you explain the reason?

---

> > > ### Author Response · Authors · 2025-04-07
> > >
> > > Thank you very much for your insightful suggestion. We fully agree that dynamically adding agents during execution is an important and realistic setting, especially in life-long or open-ended multi-agent systems. In response, we have extended our experiments to include this scenario and carefully analyzed its impact on collaborative behavior.
> > >
> > > Specifically, we designed a new experiment based on the MPE Spread task. In this setup, we fixed the number of landmarks to 4 while keeping the number of agents as 3 during training. During execution, however, we introduced an additional agent with a fixed policy to simulate the case of dynamically joining agents. The goal was to evaluate whether the original 3 agents, trained without the presence of this fourth agent, could adapt their strategies on-the-fly to maintain effective collaboration.
> > >
> > > The quantitative results of this experiment are presented in Table 5 (see [anonymous link](https://anonymous.4open.science/r/MCGD_Rebuttul-23DF)). Notably, our proposed method, MCGD, demonstrates superior collaboration robustness compared to baselines, both during training and under dynamic execution conditions. In particular, the performance gain upon adding the new agent during testing reaches 11.7%, highlighting MCGD's adaptability in dynamic multi-agent environments.
> > >
> > > To further illustrate the cooperative behavior adjustment, we visualized the agent trajectories in both 3-agent and 4-agent execution settings (see Figure 2 in the [anonymous link](https://anonymous.4open.science/r/MCGD_Rebuttul-23DF)). During training, due to the mismatch between the number of agents and landmarks, agents developed strategies that did not rely on strict 1-to-1 assignment. For instance, some agents learned to minimize the combined distance to two landmarks rather than commit to a single target.
> > >
> > > In the dynamic execution phase, the newly added agent initially starts far from all landmarks, and thus the original 3 agents continue their learned behavior. However, as the new agent approaches a specific landmark, the other agents dynamically revise their goals, often yielding their original targets and reassigning themselves to the closest remaining landmarks. This behavior shift results in a significant performance boost and offers a clear demonstration of MCGD's robust cooperation under dynamic agent populations.

---

### Official Review · Reviewer_PSeB · 2025-03-12

**Overall Recommendation:** 3

**Summary:**

This paper introduces Multi-agent Coordination based on Graph Diffusion (MCGD), a novel framework for offline multi-agent reinforcement learning (MARL) that uses graph diffusion models to enhance coordination in dynamic environments. MCGD constructs a coordination graph to capture multi-agent interactions and uses a form of categorical and anisotropic diffusion processes to model agent interactions and actions. The framework outperforms existing state-of-the-art baselines in coordination performance and policy robustness across various multi-agent environments.

## update after rebuttal
I appreciate the additional experiments and hope any remaining corrections are fixed in the final version of the paper.

**Claims And Evidence:**

The major claims are supported by results on a set of multi-agent benchmarks with off-the-shelf datasets. The graph diffusion model for multi-agent coordination using a graph transformer network appears novel.

However, the claim that the anisotropic diffusion process models the diversity in single agent actions is not adequately explored. “Diversity” could be quantified better (such as a metric based on mutual information [1] or SND [2]) and supported with experiments.

**Essential References Not Discussed:**

References used throughout the review are shown below.

References:

[1] Celebrating Diversity in Shared Multi-Agent Reinforcement Learning, Li et al, NeurIPS 2021

[2] Controlling Behavioral Diversity in Multi-Agent Reinforcement Learning, Bettini et al, ICML 2024

[3] Discrete GCBF Proximal Policy Optimization for Multi-agent Safe Optimal Control, Zhang et al, ICLR 2025

[4] Scaling Safe Multi-Agent Control for Signal Temporal Logic Specifications, Eappen et al, CoRL 2024

[5] Graph Convolutional Reinforcement Learning, Jiang et al, ICLR 2020

[6] Graph Policy Gradients for Large Scale Robot Control, Khan et al, CoRL 2020

**Experimental Designs Or Analyses:**

I did not find any issues in the experiments following prior Offline MARL approaches and the ablation study on the categorical and anisotropic diffusion. The claims of capturing agent diversity could be better quantified or supported with evidence.

**Methods And Evaluation Criteria:**

For categorical noising, a transition matrix (Eq. 6) is derived from the cosine similarity between agent observations. The intuition behind this formula is not readily apparent. An explanation could be a high similarity between agent observations implies a high value at $Q_{ij}$ meaning the agents are connected. The authors could further address the motivations of this transition matrix.

The continuous node attributes $A_t$ in graph $G_t = (A_t, E_t)$ are $d$ dimensional and encode the agent actions. It is unclear if this means it stores the actions of all agents (with a common dimension $d$) or is some form of action embedding. The method to extract an action for each agent from the predicted $\hat{A}^t$ is not obvious.

The Q-loss used in the anisotropic diffusion loss (Eq. 15) is not explained in the main body. For instance, how are the Q-values estimated? Is this estimated from the offline data directly using a standard Bellman error objective? Additionally, the incorporation of average agent observations into the Q-value is not a common practice (to the best of my knowledge) and warrants further explanation.

Lastly, the ground truth $E$ (edge matrix) which is used to train the denoiser is the nearest-neighbor graph. It would help to explain how the final denoising network generated something better than just using the nearest neighbors (as evident in Fig.5, the MCGD-AD baseline).

**Other Comments Or Suggestions:**

- The usage of subscripts and superscripts of $t$ is inconsistent (e.g., Pg6 L312 uses a superscript $t$ unlike earlier) and should be fixed.
- Fig. 4 would be better understood if the trajectory states were faded based on time (solid at end of trajectory, faded at beginning).
- If Fig. 1 it would be good to depict the missing ninth agent  (there are nine on the left but only eight on the right).
- Some references need to be fixed with the year:
    - Shi, D., Tong, Y., Zhou, Z., Xu, K., Wang, Z., and Ye, J. Graph-constrained diffusion for end-to-end path planning
    - Trippe, B. L., Yim, J., Tischer, D., Baker, D., Broderick, T., Barzilay, R., and Jaakkola, T. S. Diffusion probabilistic modeling of protein backbones in 3d for the motif-scaffolding problem.
    - Vignac, C., Krawczuk, I., Siraudin, A., Wang, B., Cevher, V., and Frossard, P. Digress: Discrete denoising diffusion for graph generation
- For the following, the latest reference is ICLR 2023 :

    Wang, Z., Hunt, J. J., and Zhou, M. Diffusion policies as an expressive policy class for offline reinforcement learning

**Other Strengths And Weaknesses:**

N/A

**Questions For Authors:**

1. What is the intuition behind Eq. 6 and the use of cosine similarity?
2. Could there be added background on the Q-loss used in the anisotropic diffusion loss?
3. What other works consider the incorporation of average neighborhood agent observations into the Q-value loss?

**Relation To Broader Scientific Literature:**

This paper directly addresses the Offline MARL setting like MADIFF (Zhu et al) and methods like OMAR (Pan et al). The idea of capturing agent interactions in multi-agent systems using a graph has been examined previously [3-6] but not via diffusion of the interaction graphs (to the best of my knowledge).

**Theoretical Claims:**

Theorem 4.1 is sound.

---

> ### Author Rebuttal · Authors · 2025-03-31
>
> We sincerely thank the reviewer for the constructive feedback. We have addressed all comments and revised the manuscript accordingly. Responses are organized by reviewer Weaknesses (W) and Questions (Q), with relevant figures and tables provided in the [anonymized supplementary material](https://anonymous.4open.science/r/MCGD_Rebuttul-23DF).
>
> $\bullet$ W1: Diversity Metrics in Anisotropic Diffusion.
> As mutual information-based metrics [1] require costly conditional entropy estimation over agent identity, we adopt System Neural Diversity (SND) [2] to measure action diversity.
> On SMAC tasks, we sample $N_1$ observations and generate $N_2$ actions per agent. SND is then estimated using Sinkhorn divergence over pairwise action distances.
> As shown in Table 3 (anonymized supplementary), MCGD consistently outperforms MADIFF and DOM2, especially in scenarios with more agents and heterogeneous unit types, validating its effectiveness in modeling diverse coordination.
>
> $\bullet$ W2 and Q1: Intuition behind Transition Matrix.
> The matrix $Q$ is based on cosine similarity between agent observations, indicating that agents in similar states are more likely to substitute each other in coordination roles; that is, a higher $Q_{ij}$ implies agent $n_j$ can replace $n_i$.
> The formulation of $Q$ follows prior categorical diffusion work [3].
> While we use cosine similarity for simplicity, our framework supports alternative metrics, ensuring adaptability across environments.
>
> $\bullet$ W3: Continuous Action Attribute.
> In continuous action spaces, the matrix $A_t$ stacks raw actions (dimension $d$) for all agents.
> At inference, each agent $n_i$ retrieves the $i$-th row of $\hat{A}_t$ as its action, enabling decentralized execution.
>
> $\bullet$ W4, Q2, and Q3: Q-loss in Anisotropic Diffusion.
> Following [4, 5], we add a conservative Q-loss to complement the surrogate loss [6] in offline RL. The full training objective is provided in Equation 1 (anonymous link).
>
> The term $\overline{o}^t_i$ in the Q-value denotes the mean-pooled encoding of neighboring observations, which reduces parameters and enhances generalization when local similarity holds. As shown in the ablation study on observation processing (Table 2, anonymous link), MCGD-AO (average observation) outperforms MCGD-FC (feature concatenation) in both performance and efficiency, validating this design.
>
> $\bullet$ W5: Generated Coordination Graph.
> Though trained with nearest-neighbor (NN) graphs as supervision, our graph diffusion model adaptively predicts more informative coordination structures.
> As shown in Figure 1 (anonymized supplementary), the denoised graph evolves with agent dynamics—remaining sparse when agents are far apart, and gradually forming structured coordination as they converge. In modified scenarios, the model shifts focus to active agents, deviating from the static NN pattern.
>
> $\bullet$ W6: Capturing Agent Interaction Using Graph Structure.
> While prior methods [7–10] have applied interaction graphs in MARL, our work introduces the first graph diffusion-based framework that jointly models structural and action diversity in offline settings.
>
> Building on observation-based heuristics [7], we employ categorical diffusion for edge dynamics and anisotropic diffusion for continuous actions, enabling behavior-adaptive coordination. We plan to explore the integration of advanced graph learning techniques [8–10] in future work.
>
> $\bullet$ W7: Inconsistency between Subscripts and Superscripts.
> We have reviewed the manuscript and corrected all inconsistencies.
>
> $\bullet$ W8: Adjusting for Figure 1 and Figure 4.
> Figure 4 has been updated with a fading color scheme to better illustrate temporal progression.
> In Figure 1, the right subplot depicts a vehicle going offline and remaining stationary, overlapping with its initial position.
>
> $\bullet$ W9: Updated Reference.
> We have corrected the references to ensure the years and versions are up-to-date.
>
> References:
>
> [1] Celebrating diversity in shared multi-agent reinforcement learning, Li et al, NeurIPS 2021.
>
> [2] Controlling Behavioral Diversity in Multi-Agent Reinforcement Learning, Bettini at al, ICML 2024.
>
> [3] Graph-Constrained Diffusion for End-to-end Path Planning, Shi et al, ICLR 2024.
>
> [4] Diffusion Policies as an Expressive Policy Class for Offline Reinforcement Learning, Wang et al, ICLR 2023.
>
> [5] Beyond Conservatism: Diffusion Policies in Offline Multi-agent Reinforcement Learning, Li et
> al, CoRR 2023.
>
> [6] Dpm-solver: A fast ode solver for diffusion probabilistic model sampling in around 10 steps, Lu et al, NeurIPS 2022.
>
> [7] Graph Convolutional Reinforcement Learning, Jiang et al, ICLR 2020.
>
> [8] Discrete GCBF Proximal Policy Optimization for Multi-agent Safe Optimal Control, Zhang et al, ICLR 2025.
>
> [9] Scaling Safe Multi-Agent Control for Signal Temporal Logic Specifications, Eappen et al, CoRL 2024.
>
> [10] Graph Policy Gradients for Large Scale Robot Control, Khan et al, CoRL 2020.

---

### Official Review · Reviewer_YBYW · 2025-03-12

**Overall Recommendation:** 3

**Summary:**

This paper introduces MCGD, the first offline MARL algorithm based on graph diffusion models. MCGD employs a discrete diffusion process on graphs to model cooperative relationships among agents, while using a continuous anisotropic diffusion process to model each agent’s action distribution. The authors claim that MCGD can better model the dynamic interactions between agents. The effectiveness of MCGD is validated on multiple offline MARL datasets, with ablation studies conducted for both diffusion processes. Notably, MCGD demonstrates strong robustness, particularly in scenarios where agent attributes and interaction patterns undergo sudden changes.

**Claims And Evidence:**

The claims regarding the effectiveness of the proposed algorithm are well-supported by strong experimental results.

**Essential References Not Discussed:**

I have not found any.

**Experimental Designs Or Analyses:**

I have not verified the validity of the experimental results, as the authors have not provided the code, and some details remain unclear to me.

**Methods And Evaluation Criteria:**

The authors validate their approach with commonly used datasets. The proposed graph-based diffusion model is intuitively reasonable, but I have some doubts regarding certain details.

**Other Comments Or Suggestions:**

1. I am particularly interested in how the learned coordination graph structure evolves during task execution. It would be helpful if this could be illustrated in a case study similar to Figure 4.

2. The color saturation in Figure 5 is too high.

**Other Strengths And Weaknesses:**

### Strengths

1. The experimental results of MCGD are impressive.

2. The algorithm proposed in the paper is novel to me, and the motivation is intuitive.

### Weaknesses

1. Some descriptions in the paper are unclear, making it difficult to understand certain technical details. During sampling, the authors mention using the Q-function to select the optimal action (Line 5 in Algorithm 1). This raises several questions: 1) In a continuous action space, how is the optimal action selected from the Q-function? 2) The Q-function takes as input the average observation of neighboring agents—how is this obtained during testing when other agents’ observations are not directly available? 3) For discrete action spaces, how does the Gaussian diffusion process generate the node action attributes?

2. Additionally, is the node attribute $A_t$ the same as the joint action of the agents? If so, what is the rationale behind defining forward noising in anisotropic noising using the covariance matrix of the agent's action?

3. Some algorithm designs appear rather arbitrary and lack generality. The authors propose using the similarity in the raw observation space as a measure of cooperation between agents. However, observation similarity does not necessarily imply suitability for collaboration, as some tasks may require cooperation between agents with different characteristics. Moreover, such similarity is influenced by the specific meaning of each dimension in the observation space, which varies across different environments. While this property may hold in the environments tested by the authors, it is difficult to claim general applicability. Additionally, using the average observation of neighboring agents as input to the Q-function also lacks generality. Directly averaging observations can lead to significant information loss; for example, if two neighboring agents have values of 0.5 and 0.5 in a certain dimension, their average would be indistinguishable from another pair with values of 0.9 and 0.1, despite the differences in underlying distributions.

**Questions For Authors:**

See weaknesses. If the authors can address my concerns, I am willing to increase my rating.

**Relation To Broader Scientific Literature:**

The proposed algorithm builds upon works in graph diffusion and offline MARL.

**Theoretical Claims:**

No, I did not check the details of the proof.

---

> ### Author Rebuttal · Authors · 2025-03-31
>
> We sincerely thank the reviewer for the constructive feedback. We have addressed all comments and revised the manuscript accordingly. Responses are organized by reviewer Weaknesses (W) and Questions (Q), with relevant figures and tables provided in the [anonymized supplementary material](https://anonymous.4open.science/r/MCGD_Rebuttul-23DF).
>
> $\bullet$ W1.1: Action Selection in Continuous Space.
> In the policy sampling phase, each agent $n_i$ generates $N$ random actions from the continuous space to form a candidate set, which is evaluated by the trained Q-function $\mathcal{Q}_{\phi_i}$ to select the action with the highest Q-value.
> This replaces the Gaussian noise initialization used in prior methods [1,2], offering a more value-guided and sample-efficient strategy. As selection is over a finite candidate set, the approach naturally applies to both discrete and continuous spaces without requiring action differentiability or closed-form maximization.
>
> $\bullet$ W1.2 and W3.2: Explanation of Average Observation.
> To reduce model size and improve scalability, we apply a shared MLP to each neighboring observation and use mean pooling over the extracted features. Compared to concatenation, this approach is more parameter-efficient and robust, particularly as the number of neighbors increases. By focusing on similar neighbors, it also mitigates potential information loss.
>
> Ablation results on SMAC (Table 2 in anonymous link) show that MCGD-AO (average observation) outperforms MCGD-FC (feature concatenation) in both performance and efficiency, validating our design.
>
> During testing, due to decentralized constraints, agents substitute their own observation for the averaged neighbor input, consistent with standard MARL practices.
>
> $\bullet$ W1.3: Gaussian Diffusion over Discrete Actions.
> In discrete action spaces, we use one-hot encoding to represent actions and apply softmax decoding at the diffusion model's output. This embeds discrete actions into a continuous latent space for Gaussian diffusion, while allowing valid discrete action reconstruction via softmax followed by argmax during inference.
>
> $\bullet$ W2: Rationale of Covariance Matrix.
> The node attribute $A_t$ represents the joint action across all agents in the coordination graph. To enhance coordination modeling, we extend prior works [1–3] by introducing an adaptive covariance matrix in the anisotropic diffusion to capture action uncertainty while preserving the collaboration structure. Inspired by [4], we modify only the covariance (not the mean), avoiding training instability and ensuring convergence and computational efficiency during diffusion.
>
> $\bullet$ W3.1: Observation Similarity for Collaboration.
> Our method leverages observation similarity in both initializing the dynamic coordination graph and defining the transition matrix in categorical diffusion.
>
> For initialization, we follow prior work [5] that uses observation similarity to form initial neighbor sets. This provides a flexible starting point, while the diffusion process adaptively refines the graph, enabling coordination even among heterogeneous agents.
>
> In categorical diffusion, cosine similarity between observations defines edge transition probabilities, capturing substitutable coordination. This aligns with adaptive categorical diffusion designs [6]. Other similarity metrics scaled to [0,1] can also be used, ensuring flexibility across environments and observation modalities.
>
> $\bullet$ W4: Learned Coordination Graph.
> We illustrate the evolution of the coordination graph in Figure 1 (anonymized supplementary), using the MPE Spread task. The x-axis denotes timesteps, and the y-axis represents different settings.
>
> Initially, the diffusion process disrupts edges due to large agent distances, resulting in independent behavior. As agents converge, the graph recovers a structured form, enabling coordinated behaviors such as landmark assignment.
>
> In modified scenarios, edges shift toward active agents, with Agent 0 either delayed in coordination or excluded entirely. These cases highlight the model’s ability to adapt the graph structure based on real-time agent dynamics.
>
> $\bullet$ W5: Adjusting for Figure 5.
> We have reduced the color saturation in Figure 5 to enhance visual clarity.
>
> References:
>
> [1] Beyond Conservatism: Diffusion Policies in Offline Multi-agent Reinforcement Learning, Li et al, CoRR 2023.
>
> [2] Madiff: Offline multi-agent learning with diffusion models, Zhu et al, NeurIPS 2024.
>
> [3] Diffusion-based Episodes Augmentation for Offline Multi-Agent Reinforcement Learning, Oh et al, ICML 2024.
>
> [4] Directional diffusion models for graph representation learning, Yang et al, NeurIPS 2023.
>
> [5] Graph Convolutional Reinforcement Learning, Jiang et al, ICLR 2020.
>
> [6] Graph-Constrained Diffusion for End-to-end Path Planning, Shi et al, ICLR 2024.

---

> > ### Comment · Reviewer_YBYW · 2025-04-09
> >
> > I appreciate the authors' clarification of my questions, as well as the additional experiments and visualizations. The proposed algorithm is relatively complex, with many details requiring clearer explanation for the reader to fully understand. I recommend that the authors substantially revise the methodology section to include more detailed descriptions where necessary. Considering the strong empirical performance and the results of the additional experiments, I am changing my rating to weak accept.

---

### Official Review · Reviewer_PGtc · 2025-03-15

**Overall Recommendation:** 4

**Summary:**

This paper uses a graph diffusion approach to study MARL problems. This method incorporates graph diffusion in order to incorporate changes in multi-agent coordination dynamics (such as an agent dropping out). The goal of the approach is to be able to more seamlessly handle out-of-distribution states and actions than alternative MARL approaches. Experimental results indicate that the method performs well empirically.

**Claims And Evidence:**

Yes, it seems that the authors have presented experimental results that support the claims with empirical evidence.

**Essential References Not Discussed:**

I am unsure of whether references are complete, but it seems the authors made a solid effort.

**Experimental Designs Or Analyses:**

It seems that most of the MARL settings considered have relatively few number of agents (I believe less than 10 agents, is that correct)? I'm curious how well the method scales to more agents, especially given the computational costs of diffusion models in general.

**Methods And Evaluation Criteria:**

I am not an expert in applied MARL research, so I am not sure what is standard, but the examples seemed logic and reasonable to me.

**Other Comments Or Suggestions:**

Please see questions below.

**Other Strengths And Weaknesses:**

The paper is overall well written and it seems the authors have made an effort to provide a detailed explanation of the forward and backward denoising process.

**Questions For Authors:**

(1) How well can the method scale and how expensive is it compared to the other baselines?

**Relation To Broader Scientific Literature:**

I am unsure of the state of the art as I am not an expert in this domain.

**Theoretical Claims:**

I skimmed the proof and it seemed reasonable.

---

> ### Author Rebuttal · Authors · 2025-03-31
>
> We sincerely thank the reviewer for the constructive feedback. We have addressed all comments and revised the manuscript accordingly. Responses are organized by reviewer Weaknesses (W) and Questions (Q), with relevant figures and tables provided in the [anonymized supplementary material](https://anonymous.4open.science/r/MCGD_Rebuttul-23DF).
>
> $\bullet$ W1 and Q1: Scalability Comparison.
> To ensure fair and comparable evaluation, we adopt similar experimental settings to prior diffusion-based offline MARL works [1-3], where most tasks involve fewer than 10 agents. However, this does not suggest our method is limited to small-scale scenarios.
>
> As detailed in Appendix 7.4.2, we compare the policy sampling time of our method with MADIFF, showing that despite the additional cost from structural diffusion, our approach remains computationally competitive.
>
> To assess scalability and collaborative performance in larger-scale settings, we further conduct experiments on the MPE Spread task with 8, 16, 32, and 64 agents. As presented in Table 1 of the anonymized supplementary material, MCGD consistently outperforms MADIFF and DOM2 across all scales, with performance gains increasing as the number of agents grows. These results demonstrate that our graph diffusion-based design not only scales well but is also more effective in modeling complex multi-agent coordination.
>
> References:
>
> [1] Beyond Conservatism: Diffusion Policies in Offline Multi-agent Reinforcement Learning, Li et al, CoRR 2023.
>
> [2] Madiff: Offline multi-agent learning with diffusion models, Zhu et al, NeurIPS 2024.
>
> [3] Diffusion-based Episodes Augmentation for Offline Multi-Agent Reinforcement Learning, Oh et al, ICML 2024.

---

### Decision · Program_Chairs · 2025-05-01

**Decision:**

Accept (spotlight poster)

**Comment:**

This paper presents MCGD, a novel offline multi-agent reinforcement learning (MARL) framework leveraging graph diffusion models to enhance coordination and adaptability in dynamic environments. The method combines categorical diffusion to model inter-agent relationships and anisotropic diffusion for action modeling. The work is evaluated across standard offline MARL benchmarks and extended to dynamic scenarios involving changing agent availability and population sizes.

Several key sections, such as those involving the use of Q-values, coordination graphs, and the diffusion processes were found difficult to follow in the original submission, but the authors’ response helped  clarify these in the rebuttal. Some algorithmic decisions (e.g., using observation similarity for graph construction or averaging neighbor observations) were initially questioned for their generality. The rebuttal provides solid empirical and theoretical rationale, though further elaboration in the final paper would strengthen these justifications.